# A pleurocidin analogue with greater conformational flexibility, enhanced antimicrobial potency and in vivo therapeutic efficacy

Giorgia Manzo[1,8], Charlotte K. Hind[2,8], Philip M. Ferguson[1,8], Richard T. Amison[1,3,8], Alice C. Hodgson-Casson [1], Katarzyna A. Ciazynska[1], Bethany J. Weller[1], Maria Clarke [1], Carolyn Lam[1], Rico C. H. Man[4], Blaze G. O' Shaughnessy[1,3], Melanie Clifford[2], Tam T. Bui [5], Alex F. Drake[5], R. Andrew Atkinson [5], Jenny K. W. Lam [4], Simon C. Pitchford[1,3], Clive P. Page[1,3], David A. Phoenix[6], Christian D. Lorenz [7✉], J. Mark Sutton [2✉] & A. James Mason [1✉]

Antimicrobial peptides (AMPs) are a potential alternative to classical antibiotics that are yet to achieve a therapeutic breakthrough for treatment of systemic infections. The antibacterial potency of pleurocidin, an AMP from Winter Flounder, is linked to its ability to cross bacterial plasma membranes and seek intracellular targets while also causing membrane damage. Here we describe modification strategies that generate pleurocidin analogues with substantially improved, broad spectrum, antibacterial properties, which are effective in murine models of bacterial lung infection. Increasing peptide–lipid intermolecular hydrogen bonding capabilities enhances conformational flexibility, associated with membrane translocation, but also membrane damage and potency, most notably against Gram-positive bacteria. This negates their ability to metabolically adapt to the AMP threat. An analogue comprising D-amino acids was well tolerated at an intravenous dose of 15 mg/kg and similarly effective as vancomycin in reducing EMRSA-15 lung CFU. This highlights the therapeutic potential of systemically delivered, bactericidal AMPs.

[1] Institute of Pharmaceutical Science, School of Cancer & Pharmaceutical Science, King's College London, Franklin-Wilkins Building, 150 Stamford Street, London SE1 9NH, UK. [2] Technology Development Group, National Infection Service, Public Health England, Salisbury, UK. [3] Sackler Institute of Pulmonary Pharmacology, King's College London, London, UK. [4] Department of Pharmacology and Pharmacy, LKS Faculty of Medicine, The University of Hong Kong, Hong Kong, Hong Kong. [5] Centre for Biomolecular Spectroscopy and Randall Division of Cell and Molecular Biophysics, King's College London, New Hunt's House, London SE1 1UL, UK. [6] School of Applied Science, London South Bank University, 103 Borough Road, London SE1 0AA, UK. [7] Department of Physics, King's College London, London WC2R 2LS, UK. [8] These authors contributed equally: Giorgia Manzo, Charlotte K. Hind, Philip M. Ferguson, Richard T. Amison. ✉email: chris.lorenz@kcl.ac.uk; Mark.Sutton@phe.gov.uk; james.mason@kcl.ac.uk

The 2016 review on antimicrobial resistance[1] (AMR) predicts that, unless action is taken, around 10 million deaths per year will be attributable to AMR by the year 2050. Action recommended by the review is twofold: (1) that the inappropriate use of existing antimicrobials should be reduced so that their utility endures for longer and; (2) new antimicrobials must be made available that are effective against drug-resistant bacteria. The pipeline of new antibiotics is limited however, hence the potential of numerous alternatives to antibiotics—"non-compound approaches (i.e. products other than classic antibacterial agents) that target bacteria or approaches that target the host"—is being actively investigated[2].

Commissioned by the Wellcome Trust, a pipeline portfolio review of alternatives to antibiotics recommends "strong support for funding while monitoring for breakthrough insights regarding systemic therapy" for a tier of approaches that include AMPs[2]. The review presents the prevailing wisdom that AMPs are unsuited for systemic administration as they are poorly tolerated in animal models and susceptible to degradation. This substantially limits the scope of infection settings that are tractable to AMPs and hence their future development. There is an urgent need therefore to identify AMPs that are sufficiently potent against antibiotic resistant bacteria and well tolerated in vivo such that they are effective when delivered intravenously.

AMPs are a well-studied subset of a group of peptides that contribute to innate immunity, in a diverse range of organisms, through direct antimicrobial action and/or host defence regulation[3,4]. Identified in the Winter Flounder[5], *Pleuronectes americanus*, pleurocidin is a potent AMP with broad spectrum antibacterial activity that acts by damaging the plasma membrane[6], with activity that is dependent on their ability to adopt an amphipathic α-helix conformation[7]. It is now well established, however, that many AMPs can disrupt bacterial cell metabolism, in addition to, or in place of, their well-known membrane damaging action[8]. Pleurocidin is one such AMP and previous work supports the view that its high potency, at least against Gram-negative bacteria, such as *Escherichia coli*, is linked to its ability to cross the bacterial plasma membrane and penetrate within bacteria to attack intracellular targets[9,10]. Importantly, the ordered α-helix conformation that pleurocidin adopts in many membrane mimics or models is less apparent in those models that most closely represent a Gram-negative bacterial cytoplasmic membrane[11,12]. The increased conformational flexibility, detected when pleurocidin binds to such membranes, affords greater ability to penetrate into the hydrophobic core of the lipid bilayer[12], a property that we infer is critical to the potency associated with its intracellular penetration.

Strategies that increase the conformational flexibility of pleurocidin, when binding to model membranes, are likely to substantially alter its biological properties. Here we report the results of two such strategies. First, we hypothesised that substituting less bulky and less hydrophobic alanines for valines (both Val12 and Val16; pleurocidin-VA Table 1) located near to two key glycine

residues (respectively Gly13 and Gly17) would have substantial impact on conformational flexibility in pleurocidin[12,13]. Secondly, we hypothesised that substituting arginine for each of the four lysine residues (Lys7, Lys8, Lys14, Lys18; pleurocidin-KR Table 1) would directly increase hydrogen bonding between the peptide and the lipid headgroups, shifting the balance away from intramolecular hydrogen bonding that stabilises more ordered α-helix conformations.

Having solved the structures of the new analogues, we used a combination of both time-resolved and steady-state biophysical methods to determine the impact of the modifications on the membrane interaction to determine whether the modifications enhance and/or alter the peptide properties and bactericidal mechanisms of action. We evaluated the in vitro toxicity against mammalian cells and antibacterial performance against a panel of both Gram-negative and Gram-positive bacterial pathogens for the three pleurocidin analogues, and their all D-amino acid enantiomers, in both bacteriological and mammalian cell culture media. Further, we used an NMR metabolomic approach to investigate: (1) whether epidemic methicillin-resistant *Staphylococcus aureus* (EMRSA-15 NCTC 13616) or *Pseudomonas aeruginosa* RP73 distinguish between the (D-enantiomer) analogues to understand whether altered membrane interaction properties are manifested when challenging pathogens in vitro and; (2) whether altering the metabolic strategy of the bacteria renders them more or less susceptible to D-pleurocidin or its analogues. Finally, the therapeutic ability of D-pleurocidin-KR was demonstrated in a murine model of EMRSA-15 lung infection, when delivered intravenously.

## Results

**Peptide–lipid hydrogen bonding is directly and indirectly altered in pleurocidin analogues.** In designing both in silico and in vitro experiments it is important to consider the contribution of the three histidines in pleurocidin, the charge state of which will have substantial impact on the interaction of the peptide with membranes of varying anionic charge density. According to the Gouy-Chapman model, the effective pH will be reduced further, the greater the anionic charge density at a membrane surface.

Molecular dynamics (MD) simulations were constructed using structures of pleurocidin or its analogues determined in anionic detergent micelles as a first approximation of the negatively charged and amphipathic surface of a bacteria plasma membrane (Supplementary Fig. 1 and Supplementary Table 1). Previously we ensured histidine residues were positively charged in MD simulations, based on the observations of pH-dependent disordering of mixed zwitterionic-anionic membranes that suggested at least partial protonation at neutral pH[12]. Here we investigated pH-dependent changes in conformational disorder (Supplementary Fig. 2) using far-UV circular dichroism (CD) and determined no pH dependency in anionic 1-palmitoyl-2-oleoyl-sn-glycero-3-phospho-(1'-rac-glycerol) (POPG) membranes in the range tested. Again however, in the mixed zwitterionic-anionic model membranes an increase in conformational disorder was observed on

**Table 1 Peptides sequences, biophysical characteristics and concentration of peptide necessary to start membrane activity in electrophysiology experiments.**

| Peptide | Sequence | H | $(\mu H)_\alpha$ | $(\mu H)_{3-11}$ | Peptide concentration (μM) | |
|---|---|---|---|---|---|---|
| | | | | | DPhPE/DPhPG | DPhPG |
| Pleurocidin | GWGSFFKKAAHVGKHVGKAALTHYL | 0.421 | 0.309 | 0.340 | 10 | 5 |
| Pleurocidin-KR | GWGSFF**RR**AAHVG**R**HVG**R**AALTHYL | 0.418 | 0.311 | 0.342 | 10 | 7.5 |
| Pleurocidin-VA | GWGSFFKKAAH**A**GKH**A**GKAALTHYL | 0.348 | 0.278 | 0.323 | – | 2.5 |

All peptides were amidated at the C-terminus. Hydrophobicity (H) and hydrophobic moment assuming α $(\mu H)_\alpha$ or 3–11 $(\mu H)_{3-11}$ helix secondary structure were calculated using HeliQuest[66].

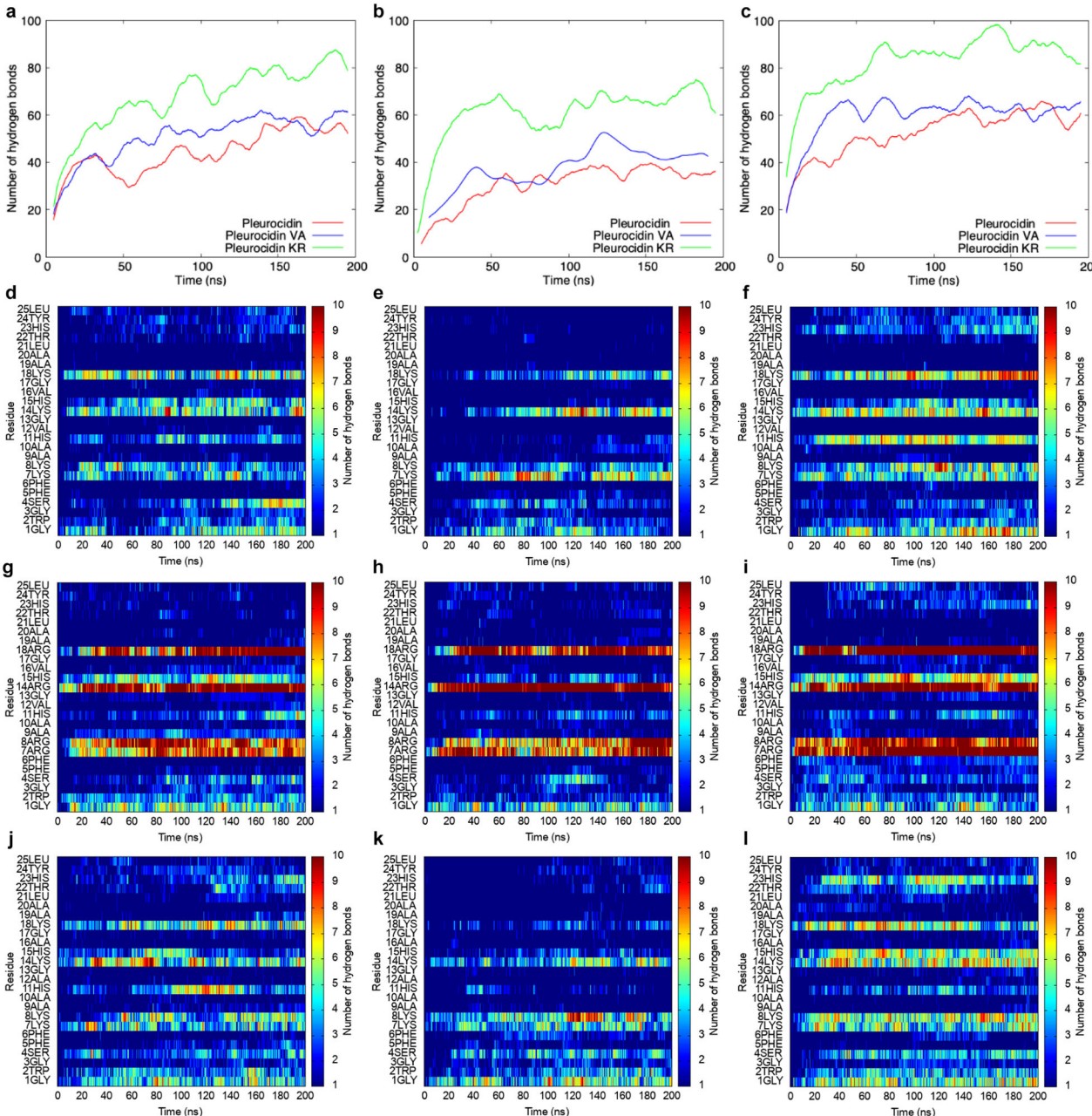

**Fig. 1 Hydrogen bonding from peptide to lipids is altered in pleurocidin analogues and affected by histidine charge state.** Total (**a–c**) or residue specific (**d–l**) peptide–lipid hydrogen bonds are shown as a function of time for pleurocidin and its two analogues in three representative MD simulations; POPE/POPG bilayers with positively charged histidines (**a**, **d**, **g**, **j**), POPE/POPG bilayers with uncharged histidines (**b**, **e**, **h**, **k**) and POPG bilayers with positively charged histidines (**c**, **f**, **i**, **l**). Residue specific data are shown for pleurocidin (**d–f**), pleurocidin-KR (**g–i**) and pleurocidin-VA (**j–l**).

lowering the bulk pH. The anionic POPG bilayers are simple models of the plasma membrane of Gram-positive bacteria while the mixed zwitterionic-anionic bilayers model the corresponding structures in Gram-negative bacteria[14,15]. Therefore, while here we have performed MD simulations both with all histidines carrying a positive or no overall charge, the simulations of pleurocidin and its analogues in anionic POPG with positively charged histidine are likely a better model of the interaction with the Gram-positive plasma membrane. In contrast, the best model of the Gram-negative plasma membrane interaction lies somewhere between the simulations with the two differing charges states.

The importance of the histidine charge state in understanding the interaction with anionic or mixed anionic/zwitterionic lipid bilayers can be seen by comparing contributions from each

residue in the peptide to hydrogen bonding to lipids (Fig. 1). In pleurocidin, His11 and His15 make important hydrogen bonding contributions to 1-palmitoyl-2-oleoyl-sn-glycero-3-phosphoetha-nolamine (POPE)/POPG bilayers but only when carrying a positive charge (Fig. 1d, e). When pleurocidin binds to a uniformly anionic bilayer surface the contribution from His11 is enhanced and a contribution from His23 is registered (Fig. 1f). Bulk pH and anionic surface charge density can therefore combine to create a selectivity switch, affecting both the probability of each of the three histidines carrying a positive charge but also the nature of the non-covalent interaction between peptide and lipids—much less peptide to lipid hydrogen bonding can be expected for pleurocidin attempting to bind to a zwitterionic surface at neutral and much more when addressing

an anionic surface in mildly acidic conditions. These two extremes, crudely, represent, respectively, the plasma membranes of a host cell and a bacterial pathogen at a site of infection[16,17].

Interestingly, pleurocidin-VA is particularly sensitive to the charge state of the three histidines as hydrogen bonding via Lys18 is also attenuated on binding to the POPE/POPG bilayer when the histidines do not carry a positive charge. His11, when positively charged, makes a more substantial contribution to hydrogen bonding than in the parent peptide (Fig. 1j, k). In contrast, when binding to the anionic POPG bilayer, hydrogen bonding via His23 is much stronger, and via His11 much weaker, than in pleurocidin (Fig. 1f, l). Substitution of Val12 and Val16 by alanine therefore has an indirect effect on peptide–lipid hydrogen bonding that is felt in segments of the peptide that are distant from the flexible region around Gly13 and Gly17. Substitution of arginines for the four lysines has a direct effect on peptide–lipid hydrogen bonding. Pleurocidin-KR forms approximately one and a half times as many hydrogen bonds with the bilayer as either pleurocidin or pleurocidin-VA (Fig. 1a, c).

Notably, when the three histidines do not carry a positive charge, the difference in total peptide to lipid hydrogen bonds is greatly enhanced (Fig. 1b) indicating that selectivity of pleurocidin-KR will be diminished. The increase in hydrogen bonding can be shown to occur almost exclusively via the four arginine residues (Fig. 1g–i) and the increase in intermolecular hydrogen bonding occurs at the expense of intramolecular hydrogen bonding that would otherwise stabilise ordered α-helix/β-turn conformation (Supplementary Fig. 3).

**Pleurocidin analogues have increased conformational flexibility in model membranes.** Although pleurocidin adopts a secondary structure with high α-helix content in many membrane mimicking environments[11,12,18,19], in those that more closely resemble the plasma membrane of a Gram-negative bacterium, i.e. rich in a mixture of zwitterionic phosphatidylethanolamine and anionic phosphatidylglycerol, the structure becomes substantially, though not completely, disordered[11,12]. The key interactions that determine the extent of this conformational disorder are unknown but, considering the impact of the lipid environment, are likely to include the sum of peptide–lipid interactions, notably hydrogen bonding as above but also hydrophobic effects, Coulombic interactions as well as the order of the lipid bilayer and the presence/absence of unsaturated acyl chains.

Both all-atom MD simulations and CD experiments show that the secondary structure of the two pleurocidin analogues differs from that of the parent molecule (Fig. 2 and Supplementary Figs. 3–6). In contrast with our recent, analogous studies of aurein 2.5 and temporin L, where both peptides adopt ordered α-helix conformations with little flexibility evident beyond the N- and C-termini[20], the present MD simulations reveal pleurocidin and its analogues exhibit substantial conformational flexibility, as evidenced by high circular variance of the psi dihedral angle, throughout the length of the peptide (Fig. 2b, e, h). The time-resolved analysis of psi dihedral angles and its circular variance can be compared with various measures of secondary structure including n–n + x hydrogen bonding, Ramachandran plots of starting and final structures, and both dictionary of secondary structure of proteins[21] and DIhedral-based Segment Identification and CLassification[22] analyses (Supplementary Figs. 3–6). Notably, while a preference for an extended α-helix conformation can be detected in pleurocidin in a long segment from Lys8 to Ala21, when eight peptides are considered (Supplementary Fig. 7), and in segments between Lys8 and Gly13 and also Ala20-Leu25, when four peptides are considered (Fig. 2b), this is largely absent from

pleurocidin-KR (Fig. 2e and Supplementary Figs. 7 and 8). Interestingly, the preference for β-turn conformation (α-helix like) at the C-terminus is retained and perhaps enhanced in pleurocidin-VA but α-helix type conformation is lost from all residues preceding Lys14 (Fig. 2h and Supplementary Figs. 7 and 8).

When the simulations are repeated with histidines carrying no overall charge, the differences in conformational preference between the three analogues are retained although the conformational flexibility of pleurocidin around the segments of α-helix conformation is notably reduced (Supplementary Fig. 8). The conformational flexibility of pleurocidin-KR and pleurocidin-VA is not noticeably affected by the protonation state of the three histidines. Finally, analogous MD simulations performed with an anionic, POPG membrane indicate that the membrane composition influences the conformation of the bound pleurocidin and its analogues (Supplementary Fig. 9). A preference for α-helix and/or type I β-turn is now displayed throughout the length of pleurocidin, a segment with dihedral angles consistent with α-helix and/or type I β-turn appears between Phe6 and His11 in pleurocidin-VA, concomitant with an increase in n–n + 4 hydrogen bonds, but pleurocidin-KR is unaffected and little or no α-helix conformation detected during the 200 ns of the duplicate simulations.

These data obtained from MD simulations of the first 200 ns of peptide binding are qualitatively supported by CD measurements performed in the steady state (Fig. 2c, f, i). This indicates that, while all three peptides have a preference for α-helix like conformation (as observed in SDS), in both POPE/POPG and POPG models membranes, the conformation of pleurocidin and, to a greater extent, pleurocidin-KR present spectra consistent with β-turn conformation (Fig. 2c, f). Pleurocidin-VA also adopts a β-turn type conformation in POPE/POPG which is greatly enhanced in POPG (Fig. 2i). The mirror image CD spectra presented by the all D-amino acid analogues indicate that D-pleurocidin and D-pleurocidin-KR adopt conformations that are the mirror image of their all L-counterparts (Supplementary Fig. 10).

**Modifications to the pleurocidin primary sequence fundamentally alter activity against model membranes.** Having established that conformation and peptide–lipid hydrogen bonding is altered in pleurocidin analogues when binding to lipid bilayers, we then examined the impact of this on bilayer penetration and disruption. In both types of membranes, but most notably in those modelling the Gram-negative plasma membrane, pleurocidin-KR inserts more readily and is much more disruptive than either its parent or the pleurocidin-VA analogue.

Consistent with previous work[12], in the 200 ns MD simulations, time-resolved insertion by all three peptides in both bilayers proceeds predominantly via the more hydrophobic N-terminus, with some penetration of the bilayer via the C-terminus segment observed, in particular in the POPG bilayers (Fig. 3a, c, e, g, i, k). The charge state of the three histidine residues is important for this process (Supplementary Fig. 11). When the histidines do not carry a positive charge, pleurocidin and its analogues struggle to insert into the bilayer as effectively, with penetration in the N-terminus restricted to the first four residues and the C-terminus failing to penetrate entirely.

Time-resolved penetration of POPE/POPG (Fig. 3a, e, i) or POPG (Fig. 3c, g, k) bilayers differed according to peptide and bilayer composition, however, with POPE/POPG bilayers providing greater discrimination. In POPG bilayers all three peptides insert into the hydrophobic core of the bilayer by the end of the 200 ns simulations with N-terminus, central and C-terminus segments all penetrating below the plane of the lipid phosphates.

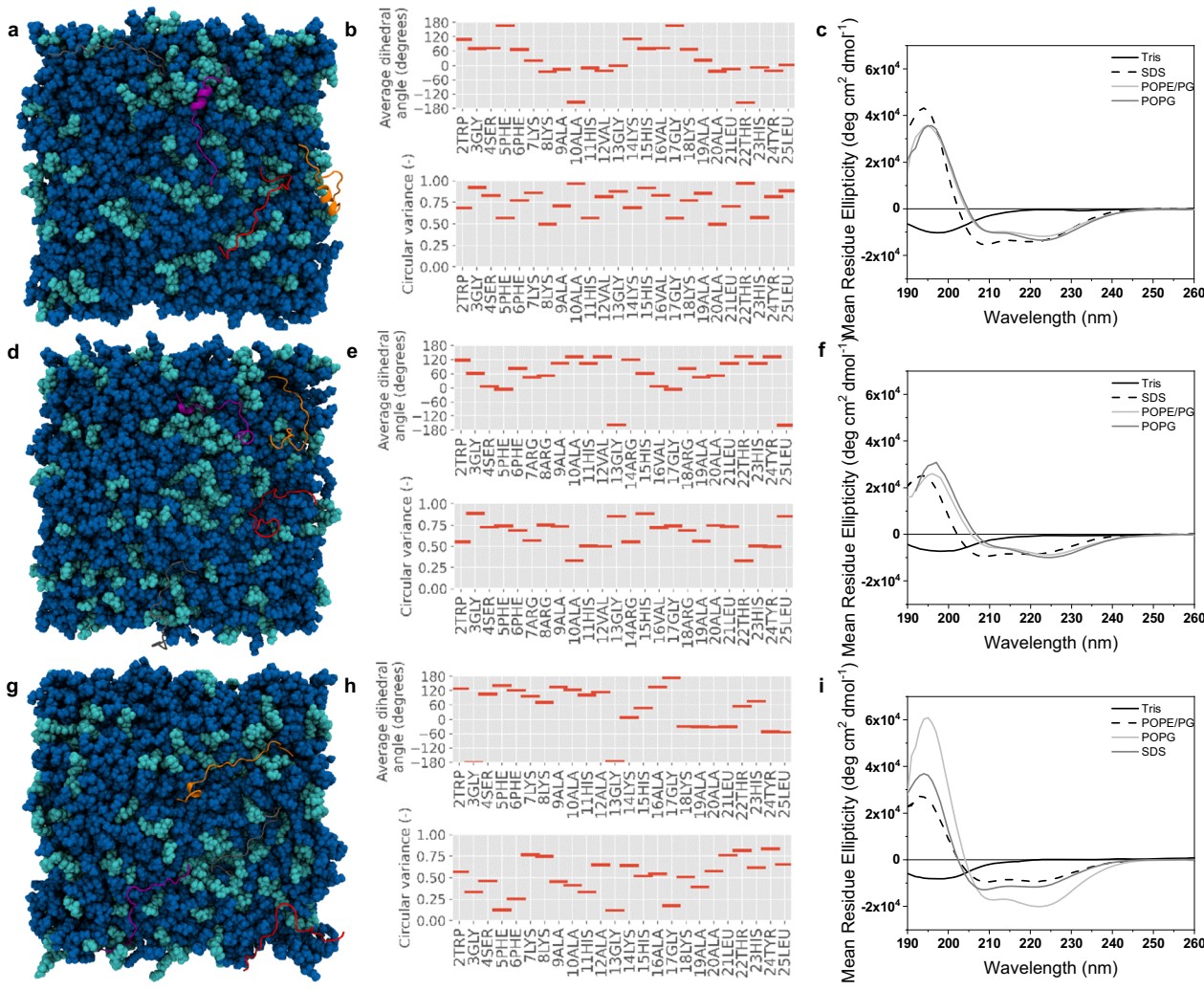

**Fig. 2 Secondary structure analysis of pleurocidin peptides.** Top view snapshots showing ordered/disordered conformation for each of eight peptide monomers (**a**, **d**, **g**) binding to POPE/POPG bilayers in silico. From the same simulations, average psi dihedral angles and circular variance of psi are shown for each residue, averaged over 200 ns of simulation and eight peptides (**b**, **e**, **h**). Far-UV CD spectra obtained in anionic SDS detergent micelles or models of Gram-negative or Gram-positive plasma membranes comprising, respectively, POPE/POPG or POPG lipids (**c**, **f**, **i**). Data are shown for pleurocidin (**a–c**), pleurocidin-KR (**d–f**) and pleurocidin-VA (**g–i**).

Some qualitative differences are apparent however with pleurocidin-KR penetrating more than pleurocidin between Gly13 and Ala19 (Fig. 3c, g) and pleurocidin-VA penetrating less than pleurocidin between Ala9 and Ala/Val12 (Fig. 3c, k).

Consistent with this, electrophysiology measurements using the Port-a-patch® automated patch-clamp system of each analogue challenging 1,2-diphytanoyl-sn-glycero-3-phospho-(1′-rac-glycerol) (DPhPG) bilayers (Fig. 3d, h, l) indicate all three peptides induce ion conductance activity at comparable peptide concentrations. Again, some qualitative differences are apparent; while all three peptides generate a mix of irregular but also channel like activities, the latter was consistently observed more frequently for pleurocidin-KR. Pleurocidin and, in particular pleurocidin-VA, were consistently more likely to trigger bursts of conductance of varying amplitude which lacked discrete opening levels.

The interaction of pleurocidin and its two analogues with POPE/POPG bilayers differs substantially. In MD simulations, much greater penetration of the hydrophobic core was observed between Lys/Arg7 and His15—the region where hydrogen bonding is enhanced by three of the four lysine to arginine substitutions—for pleurocidin-KR over pleurocidin (Fig. 3a, e).

Insertion by pleurocidin-VA is more like that of pleurocidin but penetration of Lys14/His15 is weaker and more reliant on Phe5/Phe6 at the N-terminus. Disordering of the lipid acyl chains was also monitored in the MD simulations (Fig. 3o, p and Supplementary Fig. 12) as well as by [2]H solid-state NMR of chain deuterated POPG (Fig. 3m) or POPE (Fig. 3n). In MD simulations, AMPs have been shown to disorder lipids that are close to the peptide while those that are more distant become more ordered[23,24]. Such data, from the first 200 ns of a peptide–bilayer interaction, will not perfectly correlate with [2]H NMR data obtained in the steady state. Nevertheless, both methods indicate pleurocidin-KR induces greater disorder than pleurocidin or pleurocidin-VA although manifested in the zwitterionic POPE component in the MD simulations and in the anionic POPG component by NMR (Fig. 3o, m). Electrophysiology measurements reveal both lysine to arginine and valine to alanine substitutions fundamentally alter membrane activity in 1,2-diphytanoyl-sn-glycero-3-phosphoethanolamine (DPhPE)/DPhPG models of the Gram-negative plasma membrane. Challenge with pleurocidin induces irregular and high intensity activity, with no evidence of channel like activity, that gradually subsides with the bilayer remaining intact (Fig. 3b).

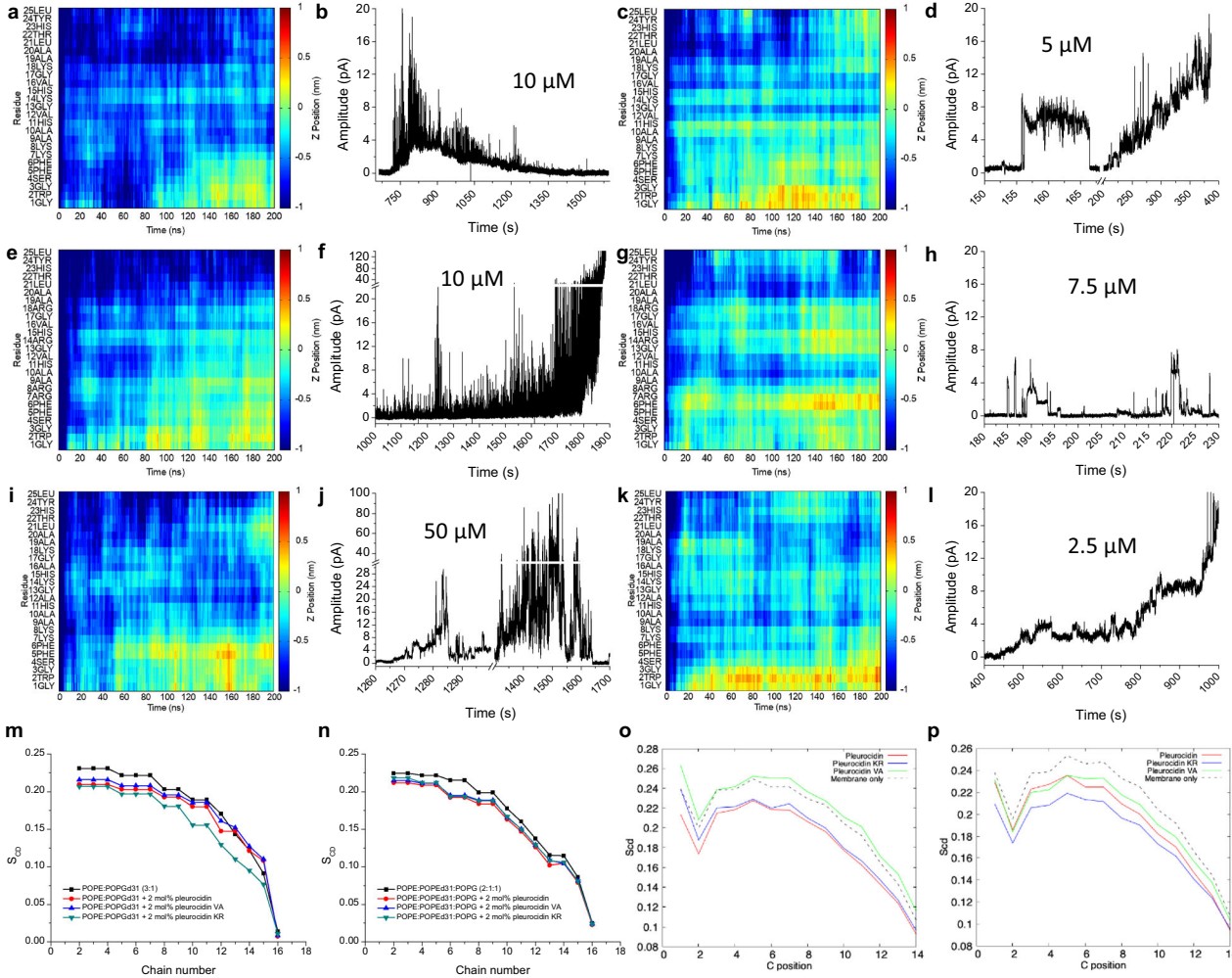

**Fig. 3 Activity of pleurocidin and its analogues on in silico and in vitro models of bacterial plasma membranes.** The depth of insertion into each membrane is shown as the Z-position for each residue, averaged over all four peptides, relative to the phosphate group plane in the upper POPE/POPG (**a**, **e**, **i**) or POPG (**c**, **g**, **k**) bilayer leaflet in six MD simulations. Positive or negative values indicate the peptides are below or above the phosphate group. Representative current traces illustrating membrane activity when DPhPE/DPhPG (**b**, **f**, **j**) or DPhPG (**d**, **h**, **l**) model membranes are challenged with each peptide at the lowest concentration that induced detectable activity. Data are for pleurocidin (**a–d**), pleurocidin-KR (**e–h**) and pleurocidin-VA (**i–l**). Membrane disordering by each peptide in mixed POPE/POPG bilayers is shown with data obtained from $^{2}$H solid-state NMR (**m**, **n**) or from MD simulations where order parameters are calculated for all lipids within 4 Å of a peptide (**o**, **p**).

This is consistent with pleurocidin inducing conductance as it passes from one leaflet of the bilayer to the other and with its known ability to penetrate within bacteria as part of its bactericidal activity. In contrast, challenge with pleurocidin-KR reproducibly leads to a gradual increase in, again, irregular conductance, until the bilayer breaks (Fig. 3f). Conductance in the DPhPE/DPhPG membrane induced by pleurocidin-VA was inconsistent. Frequently very high peptide concentrations are required to trigger activity and often no activity is detected at all. When activity is observed it is irregular and with high intensity (Fig. 3j).

**Pleurocidin analogues offer species and environmental dependent improvements in antibacterial potency, broadening the spectrum of activity.** Next we investigated whether there is evidence for the altered mechanism of action altering antibacterial activity, first through assessing potency in vitro against a panel of Gram-negative or Gram-positive pathogens using the AMPs either alone (Table 2) or in combination with clinically relevant antibiotics (Supplementary Table 2) and, second, through an NMR metabolomic study of how bacteria respond to

challenge with the all D-amino acid pleurocidin and its -KR or -VA analogues. Antibacterial susceptibility testing was conducted both in Mueller Hinton Broth (MHB), as recommended[25], and also in Roswell Park Memorial Institute 1640 (RPMI) medium. The latter was used since recent studies have shown that both false-negative[26] and false-positive[27] results can arise from exclusively testing antibiotic susceptibility in MHB alone.

In MHB, pleurocidin possesses broad spectrum activity. However, with the notable exception of *Pseudomonas aeruginosa*, activity against Gram-negative isolates is generally higher than that against Gram-positive isolates. The potency of pleurocidin-KR is marginally greater than that of pleurocidin against most Gram-negative isolates but against *P. aeruginosa* and Gram-positive isolates, the improvement is more substantial. The potency of pleurocidin-VA is comparable to the parent peptide against Gram-negative isolates but reduced against Gram-positive isolates. Most notably however, the in vitro cytotoxicity of pleurocidin-VA against four mammalian cell lines is severely attenuated (Table 2). Analogues composed entirely of D-amino acids do not gain antibacterial potency over their all L-amino acid enantiomers in MHB, suggesting there is little to no difference in their mechanism of action.

**Table 2 Antimicrobial activity and cellular toxicity.**

| Isolate | Peptide concentration (µg/ml) | | | | | |
|---|---|---|---|---|---|---|
| | Pleurocidin | D-pleurocidin | Pleurocidin-KR | D-pleurocidin-KR | Pleurocidin-VA | D-pleurocidin-VA |
| **Gram-negative** | | | | | | |
| Klebsiella pneumoniae NCTC 13368 | 4–8 (32) | 4 (8–16) | 2–4 (32) | 2 (4) | 4–8 (32) | 8 (32) |
| Klebsiella pneumoniae M6 | 4 (32) | 4 (4–8) | 2 (32) | 2 (4) | 4–8 (32) | 4 (32) |
| Acinetobacter baumannii AYE | 1–2 (4–8) | 1 (2) | 1–2 (8–16) | 2 (2) | 2 (16) | 1 (8–32) |
| Acinetobacter baumannii ATCC 17978 | 1–2 (8) | 1 (8) | 1 (8) | 2 (4) | 1–2 (32) | 1 (16) |
| Pseudomonas aeruginosa PAO1 | 64 (>32) | 2 (16) | 4 (>32) | 4 (16) | 16–32 (>32) | 4 (32) |
| Pseudomonas aeruginosa NCTC 13437 | 16–32 (>32) | 8 (16–32) | 8 (>32) | 4 (16) | 32 (>32) | 32 (>32) |
| Escherichia coli NCTC 12923 | 1–2 (16) | 1 (2–4) | 1 (16) | 1 (2) | 1 (32) | 1 (8) |
| **Gram-positive** | | | | | | |
| MS Staphylococcus aureus ATCC 9144 | 4 (**2**) | 2 (**0.5**) | 2 (4) | 2 (0.25–0.5) | 8 (32) | 4 (8) |
| EMR Staphylococcus aureus-15 NCTC 13616 | 16 (**4**) | 16 (**1**) | 4 (4) | 2 (**0.5**) | 64 (**32**) | 32 (**8**) |
| EMR Staphylococcus aureus-16 NCTC 13277 | 16 (**4**) | 16 (**1**) | 4 (8) | 4 (**1**) | 64–128 (>32) | 64 (**32**) |
| VS Enterococcus faecalis NCTC 775 | 64 (NG) | 32 (NG) | 16 (NG) | 8 (NG) | 64–128 (NG) | 64 (NG) |
| VR Enterococcus faecium NCTC 12204 | 16–32 (32) | 16 (**4**) | 4 (16) | 2 (2) | 32–64 (32) | 16 (32–64) |
| **Toxicity** | | | | | | |
| HEK293 | (87.5 ± 11.1) | (61.4 ± 3.9) | (40.1 ± 4.1) | (36.4 ± 4.3) | (~400) | (217.3 ± 46.5) |
| HeLa | (58.2 ± 3.9) | (47.1 ± 2.1) | (30.7 ± 2.6) | (25.3 ± 2.2) | (>400) | (198.2 ± 10.2) |
| A549 | n.d. | (79.3 ± 7.2) | n.d. | (56.5 ± 4.5) | n.d. | (>250) |
| Calu-3 | n.d. | (41.9 ± 1.8) | n.d. | (39.1 ± 1.8) | n.d. | (>250) |
| **SI** | | | | | | |
| HEK293 /EMRSA-15 | 5.5 (21.9) | 3.8 (61.4) | 10.0 (10.0) | 18.2 (72.8) | 6.25 (12.5) | 6.8 (27.2) |
| A549/EMRSA-15 | n.d. | 5.0 (79.3) | n.d. | 28.3 (113.0) | n.d. | >7.8 (>31.2) |

Values are given for peptides tested in Mueller Hinton broth with values obtained in RPMI given in parentheses. The selectivity index is the $EC_{50}$ divided by the MIC in the indicated conditions. Italic or bold values indicate, respectively, a significant (factor of 2 or more) reduction or improvement in potency in RPMI.

*MS* methicillin sensitive, *EMR* epidemic methicillin resistant, *VS* vancomycin sensitive, *VR* vancomycin resistant.

None of the analogues, whether comprising L- or D-amino acids demonstrate noticeable haemolysis at the concentrations where they are effective antibacterials (Supplementary Fig. 13) and the in vitro therapeutic index over red blood cell haemolysis for the majority of isolates ranges between 18- and 150-fold for D-pleurocidin-KR (10% haemolysis achieved with ~75 μg/ml).

Performing the same experiments with RPMI, supplemented with 5% fetal bovine serum (FBS), in place of MHB alters the susceptibility pattern considerably with Gram-negative and Gram-positive isolates affected differently and greater discrimination between L- and D-enantiomers and the greater proteolytic stability of the D-enantiomer is likely now important[28]. Although Gram-negative isolates are generally less susceptible in RPMI when compared with MHB, the D-enantiomers better retain their activity, with D-pleurocidin-KR notably remaining effective at 4 μg/ml or less for all but the P. aeruginosa isolates. In contrast, the potency of the pleurocidin analogues in RPMI is most commonly enhanced against Gram-positive isolates, when compared with MHB, with D-enantiomers now gaining an advantage. This improvement is most noticeable for D-pleurocidin such that its disadvantage with respect to D-pleurocidin-KR, in these conditions, is negligible. Nevertheless, the selectivity index over epithelial cells reveals D-pleurocidin-KR as the analogue with the greatest therapeutic potential for EMRSA-15, an isolate for which a murine lung infection model has been established in our laboratory.

Testing of combinations of D-pleurocidin or D-pleurocidin-KR with clinically useful antibiotics was also assessed in vitro in both MHB and RPMI (Supplementary Table 2). Both peptides act in modest synergy with the aminoglycoside tobramycin or with rifampin. This effect was most noticeable when P. aeruginosa RP73 was tested in RPMI and the same bactericidal activity is achieved with eight times less tobramycin or rifampin as is achieved without the peptide adjuvant.

While the ability of pleurocidin both to damage bacterial plasma membranes and penetrate within bacteria to access intracellular targets is established, the relative contributions of these properties to its bactericidal activity against different bacterial species may vary and may also be influenced by growth conditions. Assays with fluorescent reporter dyes have revealed that hydroxyl radical increases and membrane permeabilization following challenge with pleurocidin at its MIC vary substantially[29]. Killing of Escherichia coli ATCC 25922 or S. aureus ATCC 25923 involved substantial oxidative stress but little membrane permeabilization. In contrast, much more permeabilization of P. aeruginosa ATCC 27853 and Enterococcus faecium ATCC 19434 was induced by pleurocidin, in addition to oxidative stress. To test whether the mechanisms of action distinct from that of D-pleurocidin are adopted by its analogues and to better understand the role of bacterial metabolism in susceptibility, we performed an NMR metabolomic study (Fig. 4 and Supplementary Figs. 14–29). By culturing either EMRSA-15 or P. aeruginosa RP73 in the presence or absence of each D-pleurocidin analogue, we aim to infer differences in their bactericidal strategy from the measures the bacteria take to overcome the challenge.

In MHB, as monitored by $^1$H NMR, EMRSA-15 metabolism is a mix of fermentation, aerobic and anaerobic respiration. The production of formate, lactate and ethanol indicate threonine, serine and glycine and what little glucose is present mostly feed mixed-acid fermentative pathways, consistent with acidification of the spent media (Fig. 4a and Supplementary Figs. 14 and 16)[30,31]. Uridine consumption and uracil production are associated with peptidoglycan biosynthesis while uracil is known to be essential for anaerobic growth[30,32]. Succinate excretion to the media is consistent with anaerobic respiration, with fumarate acting as an electron acceptor.

The effect on this process, of challenging EMRSA-15 with sub-inhibitory D-pleurocidin, is profound (Supplementary Fig. 16) and results in both a fundamental change in metabolic strategy and in cellular metabolite composition (Fig. 4g and Supplementary Fig. 17). Fermentation of serine, glycine and glucose, but not threonine, is stopped following challenge with D-pleurocidin and the acidification of the spent culture is reduced (Supplementary Fig. 14). Excretion of ornithine, succinate, ethanol, lactate, leucine, methionine and 2-aminobutyrate consequently halts while consumption of uridine is also stopped. Consumption of adenosine, acetate and aspartate increases while the challenge initiates consumption of lysine, arginine, tyrosine, and valine. Lysine and valine feed into the TCA cycle via, respectively, α-ketoglutarate and succinyl-CoA. Consumption of glutamate and isoleucine from the media is reversed and these amino acids are instead excreted while formate and phenylalanine are consumed instead of being excreted. Within the cell, succinate and uracil are depleted, as are choline, citrulline and valine while acetate and betaine increase. Consumption of tyrosine and phenylalanine (via the homogentisate pathway described for Pseudomonas putida[33]) and aspartate can all be associated with the production of fumarate. Therefore, both fumarate and nitrate might be expected to be important anaerobic electron acceptors[30]. However, the halt of succinate excretion and its intracellular depletion points to a potential switch from fumarate to nitrate as an anaerobic electron acceptor. The concentration of the latter in MHB is unknown and is invisible to the NMR method, but 2′-7′dichlorodihydrofluor-escin diacetate (DCFH-DA) fluorescence, which is sensitive to reactive oxygen species and reactive nitrogen species, increases on challenge with D-pleurocidin ($p = 0.044$) but not the other analogues (Fig. 4d). Taken together, EMRSA-15 growing in the presence of D-pleurocidin is unable to use fermentation and instead relies more heavily on anaerobic respiration, while the TCA cycle is active but fed from new carbon sources.

The effects of both D-pleurocidin analogues on EMRSA-15 can be distinguished from the parent peptide (Fig. 4d, h, i). Unlike D-pleurocidin, neither D-pleurocidin-KR nor D-pleurocidin-VA affect fermentation to any great extent (Fig. 4h, i and Supplementary Fig. 16). D-pleurocidin-KR and D-pleurocidin-VA can be distinguished based on increased acetate consumption and alanine, glutamate and histidine excretion for the latter. The response of the cellular metabolite composition is more revealing, with the D-pleurocidin-VA treatment resembling more closely the effect of D-pleurocidin challenge on EMRSA-15 (Fig. 4j, l). Following challenge with sub-inhibitory D-pleurocidin-KR, eight times more potent than D-pleurocidin in MHB, there is no depletion of choline, citrulline or valine, that of succinate is mitigated and uracil is instead increased. The levels of acetate and betaine are as that observed for the unchallenged condition, but aspartate is depleted, and lactate, leucine and isoleucine levels are increased alongside plasma membrane lipid unsaturation (Fig. 4g and Supplementary Fig. 17). The mechanism of action of the more potent D-pleurocidin-KR, as inferred from changes in EMRSA-15 metabolism, is therefore fundamentally different from that of the parent D-pleurocidin in MHB and may depend on a greater contribution from membrane damage.

In glucose-rich RPMI, while ethanol is still produced, the fermentative production of lactate by EMRSA-15 is much more substantial such that the media is acidified still further, potentially triggering stress (Fig. 4b and Supplementary Fig. 14), and little or no formate is produced. Both fermentation and anaerobic respiration may again be expected with RPMI containing 2 g/litre glucose and 100 mg/litre calcium nitrate as a potential electron acceptor. As well as glucose, notably 300 mg/litre glutamine is available, and its consumption is also preferred (Fig. 4b). This might ensure that the TCA cycle is being fed via 2-

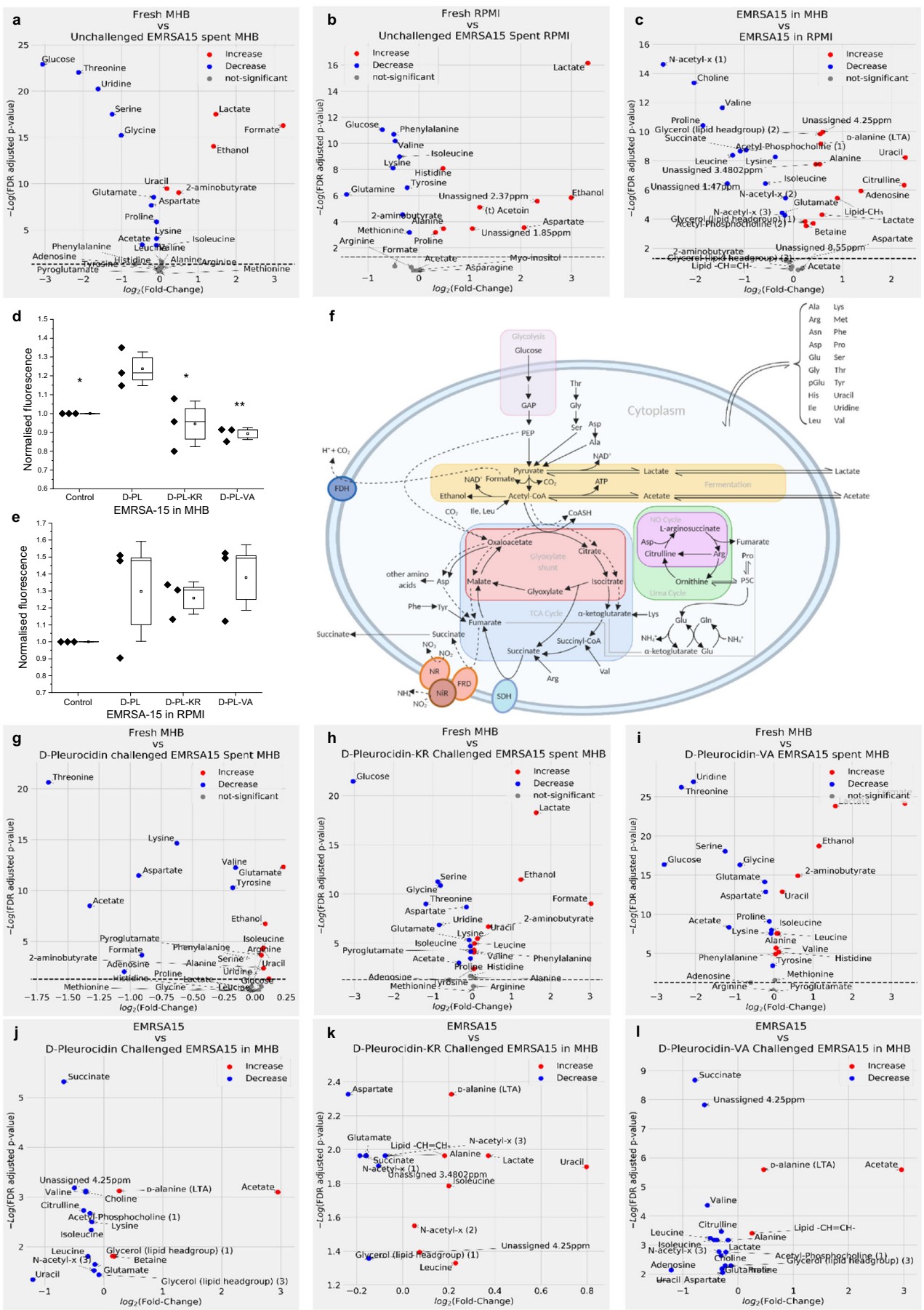

**Fig. 4 NMR metabolomics identifies altered mechanism of action in D-pleurocidin analogues.** Volcano plots obtained from liquid state $^1$H NMR of spent bacterial culture reveal the metabolic strategy of EMRSA-15 in MHB (**a**), challenged with D-pleurocidin (**g**), D-pleurocidin-KR (**h**) or D-pleurocidin-VA (**i**), or in RPMI (**b**). Tentative assignments are indicated by (t). The major catabolic pathways of EMRSA-15 are shown for context with key enzymes: FDH formate dehydrogenase, SDH succinate dehydrogenase, NR nitrate reductase, NiR nitrite reductase (**f**). $^1$H HR-MAS NMR of bacterial pellets reveals the effect on cellular metabolites of growth in these media (**c**) or challenge with antibiotics in MHB (**j–l**). Production of ROS/RNS when challenged by antibiotics, as monitored by DCFH-DA, is shown for MHB (**d**) and RPMI (**e**). Data points are averages from six distinct colonies. Boxes represent 1 SE and whiskers 1.5 SE of three independent repeats. Significance is relative to the D-pleurocidin challenge (*$p < 0.05$; **$p < 0.01$).

oxoglutarate but may also reflect the roles of glutamine as a major nitrogen donor, in osmotic protection and in multiple steps of peptidoglycan synthesis[34,35]. The effect of this change in metabolic strategy for the cellular metabolite composition is substantial (Fig. 4c) and may render the bacteria more susceptible to the action of D-pleurocidin and its analogues. In contrast with their effect in MHB, apart from a halt in proline and histidine excretion, the three analogues have only a modest impact on the metabolism of EMRSA-15 in RPMI (Supplementary Fig. 19). Although there is a trend towards an increase in DCFH-DA fluorescence on challenge with D-pleurocidin (Fig. 4e), overall the response is similar for all three peptides and, critically, glucose consumption and lactate and ethanol excretion are largely unaffected. The impact of the three analogues on the cellular metabolite composition is similar (Supplementary Fig. 20). As observed for D-pleurocidin challenge of EMRSA-15 MHB above, succinate, uracil and adenosine are depleted, and cellular acetate increases following challenge with all three analogues. Reflecting the similarity in their MIC in RPMI, apart from a modest increase in plasma membrane lipid unsaturation and a depletion of 2-aminobutyrate and lysine, there is therefore much less to distinguish the action of D-pleurocidin-KR from its parent.

A similar study was performed for *P. aeruginosa* RP73. There is little evidence of fermentation for *P. aeruginosa* RP73 when cultured in MHB or RPMI, where there is notable consumption of, respectively, formate and lactate. This strain was isolated from a cystic fibrosis patient where consumption of lactate is characteristic[36]. Lactate is a substrate for a family of lactate dehydrogenases which support anaerobic or aerobic metabolism and is present in the added FBS which is essential for growth of *P. aeruginosa* in RPMI[36]. In both media therefore a more substantial role for anaerobic respiration is expected, albeit supported by differing electron donor and acceptor pairs. Consequently, while an increase in DCFH-DA fluorescence is again detected on D-pleurocidin challenge of *P. aeruginosa* RP73 in MHB and modest changes in metabolite consumption are detected, there is no accompanying fundamental shift in metabolic strategy (Supplementary Fig. 22). Indeed, no fundamental change in metabolism is induced by challenge with either D-pleurocidin or D-pleurocidin-KR or tobramycin. In RPMI, challenge with sub-inhibitory concentrations of D-pleurocidin-KR, but not D-pleurocidin, induces a substantial increase in DCFH-DA fluorescence emission and also in the intensity of resonances assigned to lipid-CH$_3$, lipid-CH$_2$ and lipid $-CH = CH-$ while there are key differences in the metabolic perturbations it induces; most notably, isoleucine, leucine, methionine, phenylalanine, tyrosine and valine are all consumed less, relative to D-pleurocidin challenge, while arginine consumption continues and may increase, excretion of acetate substantially decreases while that of aspartate, glutamate and ornithine continues and may increase and, lysine excretion is triggered (Supplementary Figs. 23 and 27). A similar but more muted trend in membrane remodelling is observed for *P. aeruginosa* RP73 challenged in MHB (Supplementary Figs. 23 and 25). From these data, it may again be inferred that the impact of D-pleurocidin-KR is felt more keenly at the plasma membrane when compared with its D-pleurocidin parent. For *P. aeruginosa* RP73 however, while differences in

response to the two D-pleurocidin analogues can be detected, which again suggest they operate using distinct mechanisms, the effect on antibacterial potency is minor.

**D-pleurocidin-KR is an effective therapeutic in a murine model of EMRSA-15 lung infection.** Having identified that the enhanced membrane disrupting capabilities of D-pleurocidin-KR renders it less sensitive to infection setting dependent changes in bacterial metabolism and since the selectivity index for inhibition of EMRSA-15 relative to HEK293 cellular toxicity was greatest for D-pleurocidin-KR of all the analogues tested, we advanced D-pleurocidin-KR to an established murine model of EMRSA-15 infection (Fig. 5).

D-pleurocidin-KR is readily soluble in aqueous media and amenable to intravenous delivery. A dose of $1 \times 10^6$ colony forming units (CFU)/mouse of EMRSA-15 in tryptic soy agar beads inoculated in the lung establishes a stable infection ($4.26 \pm 0.24$ log$_{10}$ CFU/ml at 4 h; $4.43 \pm 0.97$ log$_{10}$ CFU/ml at 48 h) which, if untreated, causes approximately a 5% loss of weight in 48 h (Fig. 5b). Treatment with a cumulative dose of 15 or even 1.5 mg/kg/48 h D-pleurocidin-KR mitigates this weight loss to a similar extent to that achieved with 600 mg/kg vancomycin (a similar reduction may be achieved in this model with lower doses, c.f. 200 mg/kg vancomycin i.p. gives a comparable reduction in *S. aureus* USA300 CFU[37]), although the lowest dose of 0.15 mg/kg D-pleurocidin-KR has no observable effect (Fig. 5b). The two higher doses also achieve a significant reduction in CFU with 1.1 and 0.9 log$_{10}$ reduction achieved, respectively, for 15 and 1.5 mg/kg/48 h doses, comparable with that achieved with vancomycin (Fig. 5a). Since D-pleurocidin-KR was tolerated at 15 mg/kg/24 h when administered i.v., its in vivo therapeutic index is ~20-fold, before dose optimisation.

The effect of D-pleurocidin-KR therapy on the innate immune response was also characterised through analysis of bronchoalveolar lavage fluid (BAL) (Fig. 5c–i). The mouse keratinocyte chemoattractant (KC), equivalent to CXCL1 in humans, response is similar to that observed for lung CFU with infection stimulating an increase in this cytokine and treatment with vancomycin or the two higher D-pleurocidin-KR doses causing KC levels to fall to that of the uninfected group (Fig. 5h). Both IL-6 and neutrophil recruitment (reflected also in total cell numbers; Fig. 5c) are reduced following therapy with either vancomycin or D-pleurocidin-KR. However, a significant reduction is also observed for the lowest dose of D-pleurocidin-KR (Fig. 5d, f), which did not cause a significant reduction in lung CFU or mouse KC. No effects of vancomycin and D-pleurocidin-KR therapy could be detected on macrophage numbers or levels of TNFα or MCP-1 in BAL (Fig. 5e, g, i). Mouse KC is sensitive to EMRSA-15 lung infection while IL-6 reflects neutrophil recruitment and these data indicate D-pleurocidin-KR effectively reduces the burden of lung infection and may have a dampening effect on neutrophil recruitment but we do not observe evidence of a substantial innate immune response.

## Discussion
A list of 27 clinical and 9 pre-clinical studies of AMPs, published in May 2019, shows there is considerable appetite for developing

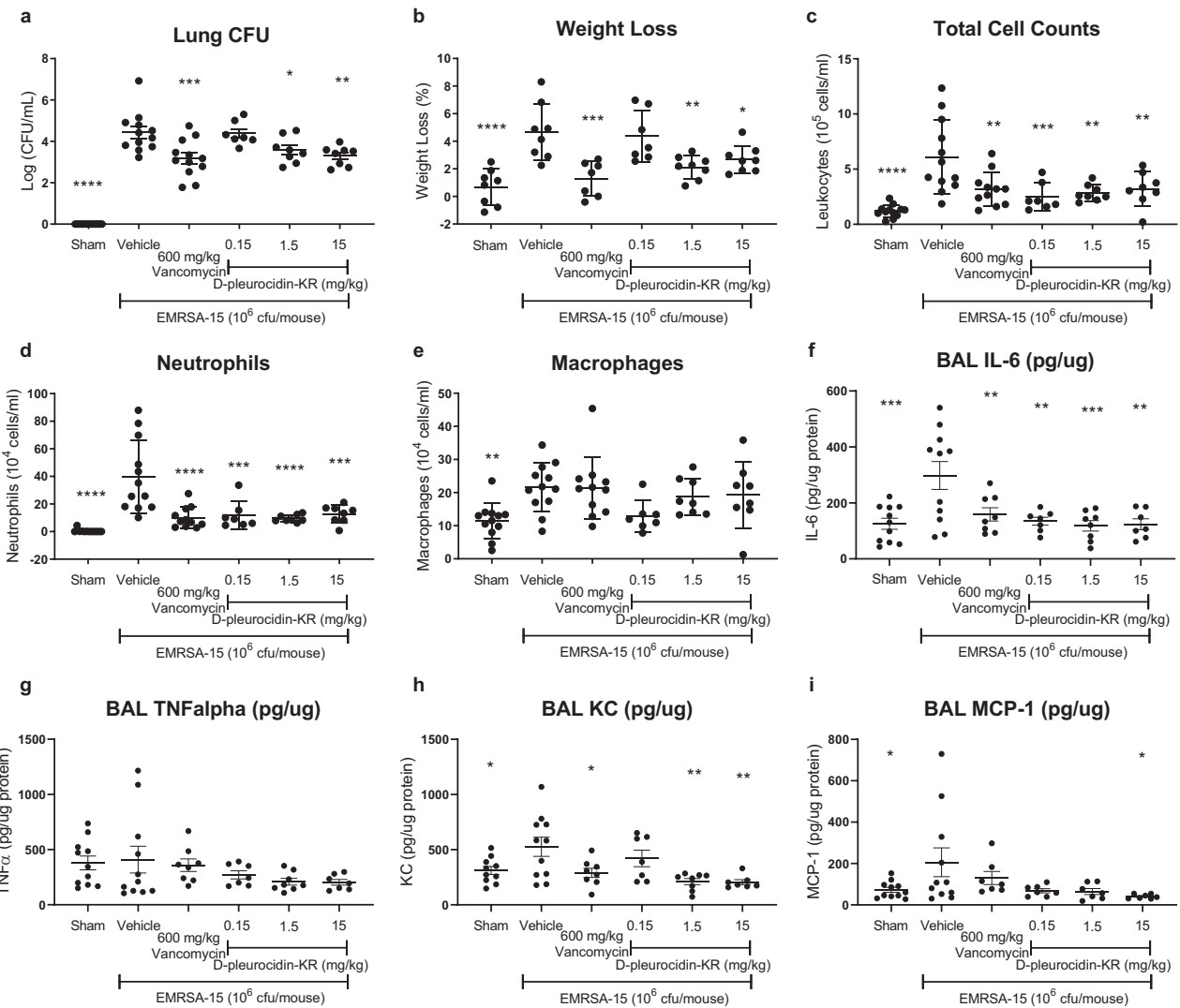

**Fig. 5 Systemically delivered D-pleurocidin-KR is effective in a murine model of EMRSA-15 lung infection.** C57Bl6J mice, challenged with $1 \times 10^6$ CFU/mouse EMRSA-15 in tryptic soy agar beads, were treated with vancomycin or D-pleurocidin-KR in three intravenous doses at 4, 24 and 30 h post infection to achieve the cumulative doses indicated. Bacterial burden in the lung (**a**), weight loss over the 48-h infection period (**b**) and BAL cells (**c–e**) and cytokines (**f–i**) reveal the effect of each intervention. Data presented are an aggregate of two independent repeats, each with control groups ($n = 6$) and peptide treatment groups ($n = 4$). Significance is indicated relative to the saline vehicle (*$p < 0.05$; **$p < 0.01$; ***$p < 0.001$; ****$p < 0.0001$). Bars represent mean and SEM.

peptide-based molecules as potential therapeutics[38]. Of these 36 AMPs however, only 7 are/were being developed for intravenous delivery. Of these seven, three have been discontinued and three (EA-230, Ghrelin and p2TA (AB103)) function via immunomodulation. Only one, hFF1-11, a lactoferricin derivative has shown direct antimicrobial activity through membrane disruption but the in vivo efficacy may also be related more to its immunomodulatory capabilities which include the release of pro-inflammatory cytokines and stimulation of monocyte differentiation[39–41]. As a result, with few if any bactericidal AMPs finding i.v. applications, a recent, authoritative review has expressed the opinion that "AMPs may never be able to achieve the same clinical outcomes as conventional antibiotics" with neither naturally occurring nor rationally designed AMPs sufficiently potent[42]. Helpfully, Haney et al identify key areas where improved understanding may nevertheless contribute to more fruitful translation of AMPs into useful therapeutic agents. These include: (1) the perception that each individual amino acid residue may have an important role and that no single active

conformation exists; (2) specific to AMPs that access the bacterial cytoplasm, better models of bacterial plasma membrane translocation concomitant with, or in the absence of, permeabilization; (3) determining what features of a given infection setting are reproduced by each in vitro assay and; (4) the concept that exogenous AMPs are unlikely to act in isolation and may act in synergy with the host innate immune system (and also clinically relevant antibiotics).

We have shown that analogues of pleurocidin, in particular D-pleurocidin-KR, are potent bactericidal AMPs which can be delivered intravenously to treat bacterial lung infections without triggering the release of pro-inflammatory cytokines or stimulating recruitment of innate immune cells in the mouse model. The modification strategy ensures that in some cases a 16-fold improvement in potency is observed for D-pleurocidin-KR over pleurocidin and this compares favourably with e.g. efforts to obtain shortened pleurocidin analogues[43,44]. Furthermore, the bactericidal activity of pleurocidin can be rendered less sensitive to changes in bacterial metabolism though manipulation of its

interaction with lipid bilayers; even relatively minor modifications are sufficient to ensure an altered mechanism of action. Both D-pleurocidin and the D-pleurocidin-KR analogue can act in synergy with the aminoglycoside tobramycin or rifampin in vitro and there is therefore some promise that AMPs may find application as adjuvants to existing, clinically relevant antibiotics. This may be achieved in particular by reducing the risk of resistance emerging, either via manipulation of seesaw mechanisms[45] or through an improved pharmacodynamics profile[46].

The MD simulation and patch-clamp studies, using bacterial plasma membrane models, offer a mutually supporting view of the variety of ways that pleurocidin AMPs may interact with the plasma membrane of differing bacteria. In the context that pleurocidin is known to both disrupt the bacterial plasma membrane but also translocates to seek intracellular targets[6,9,10], observations from patch-clamp studies identify membrane activity consistent with both of these properties. The highly irregular but high amplitude conductance that diminishes over time, observed for pleurocidin most notably in membranes that model the Gram-negative bacterial plasma membrane and remain intact, is consistent with an AMP crossing the bilayer without major structural disruption. Such activity is also observed in models of Gram-positive plasma membranes where, additionally, more channel like conductance is observed. Taken together with previous fluorescence studies that reveal species and even strain dependent differences in the extent of membrane permeabilization caused by pleurocidin at its minimum inhibitory concentration[29], this suggests that a primary mechanism of action will be to penetrate the bacteria but at higher concentrations a secondary, membrane disruptive, effect will be observed. While differences between bacterial species and strains, and also their environment will impact on which of these two effects contributes most to bacterial death, it is shown here that the interaction between peptide and lipid bilayer can be manipulated to increase the role of membrane damage in the bactericidal strategy. Notably, the MD simulations, supported by CD spectroscopy, show increasing hydrogen bonding between the bilayer and the peptide produces much greater conformational flexibility and consequently disordering of the bilayer. This highlights the importance of peptide–lipid hydrogen bonding in modulating the interaction of pleurocidin with bilayers of differing composition and shows how this, and the charge state of the three histidines, is critical in determining the outcome. Positively charged histidines enjoy favourable Coulombic interactions with anionic lipids which will also mitigate unfavourable interactions between histidines and lysines that might be positioned close in space, thus affecting not only binding to surfaces of varying anionic charge but also modulating the preferred conformation. Further, the hydrogen bonding potential of positively charged histidine is greater. This, together with varying possibilities for hydrogen bonding for lipids common in prokaryotic (phosphatidylglycerol and phosphatidylethanolamine—a primary amine) and eukaryotic (phosphatidylcholine—a quaternary amine—is more common) plasma membranes, highlights the role of histidine as a selectivity switch, influencing the likely therapeutic index of these compounds. In the absence of histidine mediated hydrogen bonding, the interaction of pleurocidin-VA with less anionic bilayers is much weaker than that of the parent in the centre and C-terminal segments. Further manipulation of this property may offer additional means of improving selectivity.

The distinct mechanisms of action of D-pleurocidin and D-pleurocidin-KR highlight two aspects of AMP development that may be important in successful translation. If it is possible to fundamentally alter the mechanism of action within members of a structure–activity relationship series with only a few amino acid substitutions, then it is likely that exogenous AMPs will differ in

mechanism from endogenous host defence peptides (HDPs), making additive relationships less likely. Further, although exogenous AMPs may share some characteristics with endogenous HDPs, it can be expected that the extent of cross resistance will be mitigated by significant differences in mechanism of action.

There is increasing recognition that susceptibility testing in bacteriological media may both under- or over-estimate the potency of different classes of antibiotics and is unlikely to replicate the conditions encountered in vivo[26,27]. In spite of well-established protocols for susceptibility testing of antibiotics, including AMPs[25], the methodology used to determine antibacterial potency of AMPs remains highly variable[42]. The use of mammalian cell culture media has been presented as a move towards better replicating the infection setting and presents AMPs with some notable challenges including increased ionic strength and the presence of serum proteases. Altered growth conditions may also affect the production of virulence factors[47,48]. The D-enantiomers of pleurocidin and its analogues did not outperform their L-amino acid parent molecules until RPMI (5% FBS) was used in place of MHB. The need to modify linear AMPs to avoid proteolytic degradation in more challenging conditions is therefore clear.

While mammalian cell culture media may present greater challenges to AMPs, and this is notable in their activities against Gram-negative isolates, bacterial metabolic activity may also affect antibiotic outcomes[49] and it is of considerable interest that, irrespective of stereochemistry, pleurocidin analogues are more potent against many of the Gram-positive isolates than their parent molecules. Further, the difference in potency between D-pleurocidin and D-pleurocidin-KR against EMRSA-15 disappears when susceptibility testing is performed in RPMI with both peptides more potent than when evaluated in MHB. EMRSA-15 growth in MHB adapts to the presence of D-pleurocidin by shutting down fermentation and placing greater reliance on anaerobic respiration, most likely using nitrate as an electron acceptor. This response is not observed when EMRSA-15 is challenged with the more potent D-pleurocidin-KR. This can be attributed to the lower concentration of D-pleurocidin necessarily used for the challenge and/or a fundamentally altered mechanism of action. The latter is supported by evidence that EMRSA-15 responds to D-pleurocidin-KR, which carries substitutions which cause greater disruption of model bilayers, by remodelling its plasma membrane. While it is not clear whether the gain in membrane disruption in D-pleurocidin-KR occurs at the expense of the ability to penetrate the bacterial cytosol, this property appears crucial in reducing sensitivity to the bacterial metabolic strategy. The increased dependence on fermentation can be implicated in rendering EMRSA-15 more sensitive to the action of D-pleurocidin analogues. While the origin of their increased potency against EMRSA-15 in RPMI (5% FBS) is not completely clear, it is nonetheless welcome and suggests that, if EMRSA-15 adopts a fermentative metabolic strategy in a lung infection setting then, all D-pleurocidin analogues will be more potent than anticipated and there is little advantage to selecting D-pleurocidin-KR. However, although mammalian cell culture conditions have been suggested as a better means of reproducing the bacterial metabolic strategy in vivo[26,27], infection setting dependent variation in nutrient availability can be expected to affect therapeutic outcomes[31]. Indeed, although fermentable substrates, including glucose are abundant in the lung infection setting, a recent study of S. aureus gene expression during cystic fibrosis lung infections finds that expression of genes involved in fermentation and use of nitrate as an electron acceptor is low[48]. This would then predict a greater chance of therapeutic success for D-pleurocidin-KR over D-pleurocidin. The varying sensitivity of the D-pleurocidin analogues to bacterial metabolism highlights

again the importance of efforts to improve our understanding of bacterial metabolism in different infection settings[48,50]. The present study, however, suggests that including a range of conditions in in vitro susceptibility testing, that stimulate different metabolic strategies in target bacteria, may be important in selecting AMPs that are resilient and improving success of AMPs in pre-clinical studies.

The immunomodulatory abilities of HDPs are increasingly recognised as being critical to their role in the innate immune system. HDPs have been shown to be capable of inducing or modifying the production of cytokines or chemokines as well as inhibiting pro-inflammatory responses from host cells which might arise from bacterial components including lipoteichoic acid, peptidoglycan, lipopolysaccharide and bacterial DNA[51]. This has led to the design of small synthetic peptides, focusing on enhancing their immunomodulatory capability[52], or enhancing the immunomodulatory capability of bactericidal peptides identified in nature[53]. Analysis of BAL fluid shows no evidence of any increased cytokine response or recruitment of either neutrophils or macrophage following i.v. D-pleurocidin-KR administration at doses that effectively reduce EMRSA-15 load in the lung. The dampening of IL-6 levels and neutrophil recruitment without any significant reduction in lung CFU does suggest that D-pleurocidin-KR can inhibit pro-inflammatory responses either directly or indirectly. However, since D-pleurocidin-KR is a highly potent (in vitro MIC 0.5 μg/ml/0.18 μM), bactericidal AMP against EMRSA-15 and, with no evidence of an immune-stimulatory effect in vivo, its therapeutic effect must currently also be primarily ascribed to a direct effect on bacteria at the site of infection.

With appropriate modification and an understanding of the requirements for bactericidal activity that is robust in the face of different bacterial metabolism and more challenging environmental conditions, pleurocidin analogues are effective in treating an EMRSA-15 lung infection. Although pleurocidin may represent a special case, our results suggest that, despite widely held concerns, bactericidal AMPs may yet be suitable for development for intravenous delivery and systemic therapeutic applications.

## Methods

**Peptides and lipids.** Pleurocidin, pleurocidin-KR, pleurocidin-VA and their all D-amino acid analogues were purchased from Cambridge Research Biochemicals (Cleveland, UK) as desalted grade (crude). The crude peptides were further purified using water/acetonitrile gradients using a Waters SymmetryPrep C8, 7 mm, 19 × 300 mm column. All peptides were amidated at the C-terminus. The lipids POPG, 1-palmitoyl-d31-2-oleoyl-sn-glycero-3-phospho-(1′-rac-glycerol) (POPG-d31), POPE, 1-palmitoyl-d31-2-oleoyl-sn-glycero-3-phosphoethanolamine (POPE-d31), DPhPG and DPhPE were purchased from Avanti Polar Lipids, Inc. (Alabaster, AL) and used without any purification. All other reagents were used as analytical grade or better. Bacterial isolates are from a collection maintained by PHE (for antibiogram see Supplementary Table 3).

**Antibacterial activity assay.** The antibacterial activity of the peptides was assessed through a modified twofold microdilution assay with modal MICs generated from at least three biological replicate experiments[20,25]. The method broadly followed EUCAST methodology, with non-cation-adjusted Mueller Hinton or RPMI replacing cation-adjusted Mueller Hinton. Peptides and antibiotics were diluted in a twofold dilution in media down a 96 well plate. Bacteria were then added, back-diluted from an overnight culture, at a starting concentration of 5 × 10⁵ CFU/ml. Plates were incubated, static at 37 C, for 20 h and the $OD_{600}$ was determined using a Clariostar plate reader (BMG Labtech). The MIC was defined as the lowest concentration where growth was <0.1 above the background absorbance. Synergy was measured using standard microdilution checkerboard assays under the same conditions as the MICs[54]. Twofold dilution series of each peptide or antibiotic were prepared in separate 96 well plates and then combined into one before addition of bacteria. The growth/no growth interface was determined using the same definition as the MIC. The fractional inhibitory concentration was calculated from the most synergistic well on the plate for three independent repeats, and presented as the average +/− standard deviation. FIC is calculated as (MIC of compound A in combination with B/MIC of compound A alone) + (MIC of compound B in combination with A/MIC of compound B alone). MICs were

determined on the same plates as the FICs to increase reproducibility. FIC values ≤0.5 were considered strongly synergistic and, consistent with a recent re-evaluation of FIC which stresses the importance of also measuring the MIC in the same microarray plate, values of 0.5 - <1 were weakly synergistic[54].

**Cytotoxicity assay.** HeLa and HEK293 cell lines were purchased from ECACC and cultured in Eagle's Minimum Essential Media containing glutamine, supplemented with 10% FBS and 1X non-essential amino acids. A549 and Calu-3 cell lines were purchased from ATCC. A549 cells were cultured in Dulbecco's Modified Eagle Medium (DMEM) while Calu-3 cells were cultured in DMEM/F-12, both supplemented with 10% FBS and 1% antibiotic-antimycotic. All cells were maintained at 5–10% $CO_2$, 37 °C. Cytotoxicity was measured by incubating starter cultures of cells for 24 h in a 96 well plate, gently removing the supernatant and replacing it with dilutions of peptides and controls, incubating for a further 24 h and then staining of the cells with the second-generation tetrazolium dye, XTT. Cells were stained with 0.2–0.3 mg/ml XTT labelling mixture with 5 mM N-methyl dibenzopyrazine methyl sulfate, subsequently incubated for 4 h and the $IC_{50}$ was calculated from the $OD_{475}$ which was measured using a Clariostar plate reader. Data presented are averages and standard error from three biological replicate experiments.

**Haemolysis assay.** Haemolysis was tested by incubating titrations of AMPs in PBS with freshly collected human red blood cells for 1 h at 37 °C. Control wells were treated with 0.1% Triton-X-100 to ensure complete lysis, or PBS-only to represent no lysis. Samples were spun down to remove non-lysed cells and the $OD_{550}$ of the supernatant was measured using a Clariostar plate reader. The percentage of haemolysis was calculated as % haemolysis = $(A_P − A_B)/(A_C − A_B) × 100$, where $A_P$ is the absorbance value for a known peptide concentration, $A_C$ is the absorbance of the 0.1% Triton-X-100 control, $A_B$ is the absorbance of the PBS control. Data presented are averages and standard error from three biological replicate experiments.

**NMR structure determination.** The NMR samples consisted of a 0.5 mM peptide solution also containing 50 mM deuterated sodium dodecyl sulphate (SDS-d25) with 5 mM Tris(hydroxymethyl-d3)-amino-d2-methane buffer at pH 7. 10% $D_2O$ containing trimethylsilyl propanoic acid (TSP) was added for the lock signal and as internal chemical shift reference. The temperature was kept constant at 310 K during the NMR experiments. NMR spectra were acquired on a Bruker Avance 800 MHz spectrometer (Bruker, Coventry, UK) equipped with a cryoprobe. Standard Bruker TOCSY and NOESY pulse sequences were used, with water suppression using a WATERGATE 3–9–19 sequence with gradients (mlevgpph19 and noesygpph19). The ¹H 90° pulse was calibrated at 37.04 kHz. The TOCSY mixing time was 90 ms, and the mixing time for the NOESY spectra was set to 150 ms. The relaxation delay was 1 s. 2048 data points were recorded in the direct dimension, and either 256 or 512 data points in the indirect dimension. The spectra were processed using Bruker TOPSPIN. The free induction decay was multiplied by a shifted-sine² window function. After Fourier transformation, the spectra were phase corrected, a baseline correction was applied, and spectra were calibrated to the TSP signal at 0 ppm.

CARA[55] (version 1.9.1.2) and Dynamo (http://spin.niddk.nih.gov/NMRPipe/dynamo) software were used for assignments and structure calculation. Inter-proton NOEs interactions were used as distance restraints in the structure calculation. After seven iterations using the simulated annealing protocol, CARA software generated a total of 200 structures on UNIO'08 (version 1.0.4)[56] and XPLOR-NIH (version 2.40)[57,58]. The 20 lowest energy structures were selected and used to produce a final average structure. In the case of CARA generating ambiguous NOE assignments, we applied the annealing protocol in Dynamo. Unambiguous NOEs only were used in this case, after being classified as strong, medium and weak on the base of their relative intensity in the NOESY spectra. Using this classification, upper limits of 0.27, 0.33 and 0.50 nm were applied, respectively, as restraint on the corresponding inter-proton distance. One thousand structures were calculated and the 100 conformers with the lowest potential energy were selected, aligned, and the root mean square deviation of the backbone heavy atoms calculated with respect to their average structure. Solvent molecules were not included in the calculations. Structural coordinates were deposited in the Protein Data Bank (www.rcsb.org) and Biological Magnetic Resonance Bank (BMRB; www.bmrb.wisc.edu) under accession codes of 6RSF and 6RSG (PDB) and 34404 and 34405 (BRMB) for pleurocidin-KR and pleurocidin-VA, respectively. Refinement statistics are presented in the supplementary material (Supplementary Table 1).

**Molecular dynamics simulations.** Simulations were carried out on either the ARCHER Cray XC30 supercomputer, or Dell Precision quad core T3400 or T3500 workstations equipped with a 1 kW Power supply (PSU) and two NVIDA PNY GeForce GTX570 or GTX580 graphics cards using Gromacs[59]. The CHARMM36 all-atom force field was used in all simulations[60–62]. All membranes in this project contained a total of 512 lipids, composed either of POPE/POPG (75:25 mol:mol) or POPG. Eight peptides were inserted at least 30 Angstrom above the lipid bilayer in a random position and orientation, at least 20 Angstrom apart. The starting structures were taken from the NMR calculation in SDS micelles. The system was

solvated with TIP3P water and Na+ ions added to neutralise. Energy minimisation was carried out at 310 K with the Nose-Hoover thermostat using the steepest descent algorithm until the maximum force was less than 1000.0 kJ/ml/nm (~3000–4000 steps). Equilibration was carried out using the NVT ensemble for 100 ps and then the NPT ensemble for 1000 ps with position restraints on the peptides. Hydrogen-containing bond angles were constrained with the LINCS algorithm. Final simulations were run in the NPT ensemble using 2-fs intervals, with trajectories recorded every 2 ps. All simulations were run for a total of 200 ns and repeated twice, with peptides inserted at different positions and orientations, giving a total of ~4.0 μs simulation. Torsion angles are circular quantities, and the circular mean of psi or phi angles may be calculated as follows:

$$\bar{\psi} = a \tan 2 \left( \frac{1}{n} \sum_{j=1}^{n} \sin \psi_j, \frac{1}{n} \sum_{j=1}^{n} \cos \psi_j \right).$$

Similarly, the associated circular variance for psi or phi angles is calculated as follows:

$$\text{Var} (\psi) = 1 - R_{av}$$

with $R$ being given by:

$$R^2 = \left( \sum_{i=1}^{n} \cos \psi_i \right)^2 + \left( \sum_{i=1}^{n} \sin \psi_i \right)^2.$$

**Liposome preparation and circular dichroism spectroscopy**. Far-UV CD spectra of the peptides bound to small unilamellar vesicles (SUV) were obtained using a Chirascan Plus spectrometer (Applied Photophysics, Leatherhead, UK) with samples maintained at 310 K. To prepare SUV, lipid powders were solubilized in chloroform and dried under rotor-evaporation. To completely remove the organic solvent, the lipid films were left overnight under vacuum and hydrate in 5 mM Tris buffer (pH 7.0). The lipid suspension was subjected to five rapid freeze–thaw cycles for further sample homogenisation. POPE/POPG (75:25, mol:mol) and POPG SUVs were obtained by sonicating the lipid suspension on Soniprep 150 (Measuring and Scientific Equipment, London, UK) for 3 × 7 min with amplitude of six microns in the presence of ice to avoid lipid degradation. The SUVs were stored at 4 °C and used within 5 days of preparation. Far-UV CD spectra were recorded from 260 to 190 nm. SUV suspension was added to a 0.5 mm cuvette at a final concentration of 5.0 mM and then a few μl of a concentrated peptide solution were added and thoroughly mixed to give a final peptide concentration of 50 μM. The same experimental conditions were used to investigate peptide secondary structure in SDS micelles, while the SDS micelles concentration was 20 with 200 μM peptide. In processing, a spectrum of the peptide free lipid suspension or SDS solution was subtracted and Savitsky-Golay smoothing with a convolution width of 5 points applied.

**Electrophysiology experiments (Patch-clamp)**. As in our earlier work[20], lipids with diphytanoyl chains are used here to form giant unilamellar vesicles (GUV). GUVs composed of DPhPE/DPhPG (60:40, mol:mol) and DPhPG were prepared in the presence of 1 M sorbitol by the electroformation method in an indium-tin oxide coated glass chamber connected to the Nanion Vesicle Prep Pro setup (Nanion Technologies GmbH, Munich, Germany) using a peak-to-peak AC voltage of 3 V, at a frequency of 5 Hz, for 120 and 140 min, respectively, at 37 °C. Bilayers were formed by adding the GUV solution to a buffer containing 250 mM KCl, 50 mM MgCl₂ and 10 mM Hepes (pH 7.00) onto an aperture in a borosilicate chip (Port-a-Patch®; Nanion Technologies) and applying 70–90 mbar negative pressure resulting in a solvent-free membrane with a resistance in the GΩ range. After formation, a small amount of peptide stock solution (in water) was added to 50 μL of buffer solution to obtain its active concentration. All the experiments were carried on with a positive holding potential of 50 mV. The active concentration, the concentration at which the peptide first showed membrane activity, for each peptide was obtained through a titration performed in the same conditions. For all the experiments a minimum of six according repeats was done. Current traces were recorded at a sampling rate of 50 kHz using an EPC-10 amplifier from HEKA Elektronik (Lambrecht, Germany). The system was computer controlled by the PatchControl™ software (Nanion) and GePulse (Michael Pusch, Genoa, Italy). The data were filtered using the built-in Bessel filter of the EPC-10 at a cut-off frequency of 10 kHz. The experiments were performed at room temperature. Data analysis was performed with the pClamp 10 software package (Axon Instruments) with estimation of pore radii performed according to the method of Tosatto et al.[63]

**Metabolomics sample preparation**. *S. aureus* EMRSA-15 and *P. aeruginosa* RP73 were streaked on Mueller Hinton agar plates and RPMI (supplemented with 5% FBS) agar plates and incubated at 37 °C overnight or until single colonies formed. On three separate occasions, three single colonies were used to inoculate 10 ml of respective media and challenges set up in broth using sub-inhibitory concentrations of: D-pleurocidin, D-pleurocidin-VA, D-pleurocidin-KR or the antibiotic tobramycin. Cultures were incubated overnight at 37 °C without shaking and pelleted by centrifugation at 4 °C. The supernatant was removed and filtered separately for spent media analysis, the pellet was briefly washed three times with phosphate buffered saline (PBS). The pellets were frozen in liquid nitrogen and

lyophilised overnight in an Alpha 1–2 LD plus freeze dryer (Martin Christ, Germany) and stored at −20 °C until required. Samples were transferred into HR-MAS inserts for use in 4 mm Bruker MAS rotor and resuspended in 30 μl of D₂O containing 3-(Trimethylsilyl)propionic-2,2,3,3-D₄ acid sodium salt (TMSP-2,2,3,3-D₄). The removed supernatant was filtered through 0.2 μm filters and pH adjusted to within 0.3 pH units within the same sample set. The samples were supplemented with 10% D₂O containing TMSP-2,2,3,3-D₄ and loaded into 5.0 mm NMR tubes.

**HR-MAS NMR**. NMR spectra were collected at 600 MHz for ¹H on a Bruker Avance II spectrometer (Bruker Biospin, Coventry, UK) with a 4 mm HR-MAS probe at 310 K while spinning at 5 kHz. ¹H NMR spectra were acquired using a Carr-Purcell Meiboom-Gill pre-saturation (cpmgpr1d) pulse sequence with a spectrum width of 16.0 ppm and 9615 data points using 64 transients. Free induction decay was multiplied by an exponential function with 0.293 Hz line broadening. 2D ¹H–¹H correlation spectroscopy (COSY) and ¹H–¹³C Heteronuclear Single Quantum Correlation (HSQC) experiments were performed on representative datasets using standard Bruker settings.

**2,7-dichlorodihydrofluorescein diacetate (DCFH-DA) fluorescence**. EMRSA-15 and *P. aeruginosa* RP73 were cultured overnight at 37 °C in either MHB or RPMI (5% FBS) in the presence and absence of D-pleurocidin or its analogues. Samples were pelleted by centrifugation and diluted to an OD₆₀₀ of 0.6 in PBS. The cell suspension was incubated in darkness for 1 h at 37 °C with 5 μM DCFH-DA. Upon completion of incubation, the cell suspension was washed with PBS to remove excess dye and incubated in darkness at 37 °C for 5 min to equilibrate. Fluorescence spectra (515–545 nm) were obtained using a Varian Cary Eclipse fluorescence spectrometer at 37 °C with excitation at 485 nm. Statistical analysis was by one-way ANOVA with Tukey's post-hoc test.

**Liquid state NMR**. ¹H NMR spectra were acquired under automation at 298 K and 700 MHz on a Bruker Avance II 700 NMR spectrometer (Bruker Biospin, Coventry, UK) equipped with a 5 mm QCl helium-cooled cryoprobe and a cooled SampleJet sample changer. 1D CPMG-presat (cpmgpr1d) experiments were acquired with 32 transients, a spectral width of 19.8 ppm and 11,904 data points. 2D ¹H-¹H COSY and ¹H-¹³C HSQC experiments were performed on representative datasets using standard Bruker settings.

**Data processing**. The spectra were Fourier transformed automatically using standard Bruker commands and manually phased and baseline corrected in Bruker TopSpin 4.0 (Bruker Biospin, Coventry, UK). Spectral pre-processing, cross-validation and multivariate analysis were carried out using the in-house programme developed by Dr Louic Vermeer and MVAPACK, an opensource Octave library for NMR metabolomic data processing and analysis[64]. Pre-processing modifies NMR spectra and reduces the variances and influences which are not of interest and may interfere with data analysis, for example, residual water peak or noise and the TSP reference peak. Initially a principle component analysis was carried out to identify clustering spectra and potential outliers in the data, due to e.g. poor baseline. Data were then subject to probabilistic quotient normalisation before binning and alignment. Orthogonal Projections to Latent Structures Discriminant Analysis analyses were conducted to establish the quality of binary comparisons of the metabolomic data (Supplementary Table 4). Volcano plots were made using the binned data for every control sample (no AMP or tobramycin) versus challenge sample (with an AMP or tobramycin), comparing fold changes in metabolites, defined as the ratio between the control and the challenge. The Mann–Whitney U test was used to compare means and associated p values were false discovery rate adjusted using the Benjamini-Hochberg method (α = 0.05). Box plots were generated using changes in normalised intensity for each metabolite and significant differences between challenges and controls were determined using a one-way ANOVA. Volcano plots and box plots were generated using in-house software developed for Jupyter Notepad using Python 3.7.0. Metabolites were assigned using the databanks: Chenomx NMR suite software (*Chenomx Inc., Canada*), Human Metabolome Database, BMRB, *E. coli* Metabolome Database and AOCS lipid library; and a comparison of chemical shifts to the literature, which was confirmed using 2D NMR spectra. Annotated spectra can be found in the supplementary materials (Supplementary Fig. 15). Metaboanalyst and KEGG were later used to identify key pathways that may be affected by these metabolites.

**Murine EMRSA-15 lung infection model**. Using the recently developed method[65], an overnight culture of EMRSA-15 was prepared in tryptic soy broth (TSB). EMRSA-15 was cultured in TSB overnight at 37 °C, adjusted to a starting OD₆₀₀ of 0.025, and grown for additional 4 h for agar beads preparation. EMRSA-15 was embedded into agar beads by mixing the overnight culture with molten tryptic soy agar, which was then spun into warmed mineral oil. The preparation was cooled and centrifuged at 2700 × g, the remaining oil was eliminated, and the beads were washed in sterile PBS. The CFU content of the beads slurry was subsequently quantified on TSA plates and the beads slurry then diluted to 2 × 10⁷ CFU/ml in sterile PBS to deliver a final dose of 1 × 10⁶ CFU/mouse. Animal work was performed in accordance with the Animals (Scientific Procedures) Act of 1986 (United Kingdom) and 2012 revisions with local ethical approval of King's College London.

On day 0, male C57Bl6J mice (8–10 weeks, Charles River) were anaesthetised under Isoflurane and 50 μl inoculum of EMRSA-15 embedded in agar beads ($10^6$ CFU/mouse) was instilled directly into the respiratory tract via oropharyngeal dosing (o.a.). Sham control mice were inoculated with sterile PBS agar beads. Animals were treated with either Vehicle (Saline), 200 mg/kg vancomycin or 0.05, 0.5 or 5 mg/kg D-pleurocidin-KR at 4, 24 and 30 h post infection via intravenous injection, reaching a total dosage of 600 mg/kg vancomycin or 0.15, 1.5 or 15 mg/kg D-pleurocidin-KR. Body weight was measured daily, and animals monitored at regular intervals for signs of pain and distress. After 48 h of pulmonary infection, animals were terminally euthanized using 25% urethane via intra-peritoneal injection. CFUs were quantified in lung homogenates. Lung homogenates were then centrifuged at 14000 rpm for 30 min at 4 °C and the supernatants were stored at −80 °C for future analysis. For BAL fluid collection and analysis (total and differential cell count), a 22-gauge catheter was inserted into the trachea and BAL was recovered by instillation of 0.5 ml of sterile PBS three times and total cells counts were performed by adding Turks stain in a 1:1 ratio. BAL fluid was then centrifuged, and supernatants were stored at −20 °C. Total levels of TNFα, IL-6, KC and MCP-1 were quantified from BAL fluid collected from all animals and measured by ELISA according to the manufacturer protocol (R&D Systems UK). Concentrations of all mediators were then normalised to total protein content of BAL fluid quantified via bicinchoninic acid total protein assays performed to manufacturers protocols (Thermofisher Scientific). Data presented are an aggregate of two independent repeats, each with control groups ($n = 6$) and peptide treatment groups ($n = 4$). Statistical analysis was by one-way ANOVA with Dunnett's multiple comparisons test.

**Reporting summary**. Further information on research design is available in the Nature Research Reporting Summary linked to this article.

## Data availability

Supplementary Information including more extensive analysis of the MD simulation data, Circular Dichroism experiments and both liquid and HR-MAS NMR metabolomic data are available. Structural coordinates were deposited in the Protein Data Bank (www.rcsb.org) and Biological Magnetic Resonance Bank (BMRB; www.bmrb.wisc.edu) under accession codes of 6RSF and 6RSG (PDB) and 34404 and 34405 (BRMB) for pleurocidin-KR and pleurocidin-VA, respectively. In addition to the structural coordinates the datasets generated during and/or analysed during the current study are available from the corresponding author on reasonable request while the data presented in the main manuscript figures are available as Supplementary Data.

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

## Acknowledgements

NMR experiments described in this paper were carried out using the facilities of the Centre for Biomolecular Spectroscopy, King's College London, and at the MRC Biomedical NMR Centre at the Francis Crick Institute. The King's instruments were acquired with a Multi-user Equipment Grant from the Wellcome Trust and an Infrastructure Grant from the British Heart Foundation. The MRC Biomedical NMR Centre is supported by the Francis Crick Institute, which receives its core funding from Cancer Research UK (FC001029), the UK Medical Research Council (FC001029), and the Wellcome Trust (FC001029). We thank Dr Tom Frenkiel and Dr Alain Oregioni for their assistance with HR-MAS NMR experiments performed at the Francis Crick Institute. This work used the ARCHER UK National Supercomputing Service (http://www.archer.ac.uk). C.D.L. acknowledges the stimulating research environment provided by the EPSRC Centre for Doctoral Training in Cross-Disciplinary Approaches to Non-Equilibrium Systems (CANES, EP/L015854/1). PMF was supported by a Health Schools Studentship funded by the EPSRC (EP/M50788X/1). AJM and GM received funding from the MRC Proximity to Discovery: Industry Engagement Fund (MC_PC_16074) and the King's Health Partners R&D Challenge rapid fund. AJM and CL received funding from a NC3Rs Skills & Knowledge Transfer grant (NC/T001240/1). CH and MC were supported by funding from PHE Pipeline fund and latterly by PHE Grant in Aid project 109505. This work utilised NIAID's suite of pre-clinical services for maximum tolerated dose (MTD) assessment (Contract no. HHSN272201700020I/75N93019F00131) conducted by Pharmacology Discovery Services Taiwan.

## Author contributions

A.J.M. wrote the main manuscript text, prepared all figures and together with J.M.S., C.D.L., D.A.P., C.P.P. and S.C.P. designed the research. G.M., K.A.C., B.J.W. and R.A.A. performed structural NMR studies. G.M. and K.A.C. performed patch-clamp studies. G.M., P.M.F., K.A.C. and B.J.W. performed CD experiments. P.M.F. and C.D.L. performed and/or analysed atomistic simulation data. P.M.F. and A.C.H.-C. performed and analysed NMR metabolomics experiments. T.T.B. and A.F.D. assisted in analysis and interpretation of CD measurements. C.K.H., M.C., R.C.H.M., J.K.W.L. and J.M.S. performed and/or designed in vitro antibacterial and toxicity assays. R.T.A., BGO'S, S.C.P. and C.P.P. performed, analysed and/or designed in vivo lung infection model studies. All authors approved the manuscript.

## Competing interests

King's College London and the Secretary of State for Health and Social Care have filed 2 UK priority applications: GB 1917644.5 and GB 2016542.9, which claim anti-microbial peptides. The authors declare no other competing interests.
