## [Peer Review File · Communications Biology]

Reviewers' Comments:

Reviewer #1:

Remarks to the Author:

In this paper, the authors describe modification strategies that generate pleurocidin analogues with substantially improved, broad spectrum, antibacterial properties, which are effective in murine models of bacterial lung infection. This is a well-written paper which merit publication. For the benefit of the readers, however, several points need clarifying and certain statements require further justification. There are given below.

1. Many researchers devoted to shorten the length of antimicrobial peptides[1, 2]. In this paper, the length of pleurocidin analogues are the same as pleurocidin. So, what are the advantages, compared with several other short peptides.
2. HEK293 is a cell line derived from human embryonic kidney cells. Hela cell line is derived from cervical cancer cells. However, the authors are focused on the bacteria which cause the lung infection and the Murine EMRSA-15 lung infection model is used in this paper. So, maybe lung cell line is suitable in cytotoxicity assay.

[1] Zhang M, Wei W, Sun Y, Jiang X, Ying X, Tao R, et al. Pleurocidin congeners demonstrate activity against Streptococcus and low toxicity on gingival fibroblasts. Arch Oral Biol. 2016;70:79-87.

[2] Souza AL, Diaz-Dellavalle P, Cabrera A, Larranaga P, Dalla-Rizza M, De-Simone SG. Antimicrobial activity of pleurocidin is retained in Plc-2, a C-terminal 12-amino acid fragment. Peptides. 2013;45:78-84.

Reviewer #2:

Remarks to the Author:

This article describes the design, synthesis and characterisation of antimicrobial peptides (AMPs), analogues of the linear basic peptide pleurocidin.

The introduction and background is well written and clear. As the authors indicate, AMPs have been the subject of intense research over the past 30 years, yet none have successfully navigated a pathway to approval for systemic use as an antibiotic.

The authors rely on extensive membrane-based molecular dynamics simulations of two modified pleurocidin derivatives – one where Val residues have been modified to alanines to increase flexibility, and the other where Lys residues are altered to Arg. They also prepared all-D amino acid versions of the peptides, designed to improve proteolytic stability. However, one of the key conclusions from these studies is that the protonation state of the His residues plays a key role in how they act. It is surprising that these residues were not also replaced e.g. with permanently charged residues, to validate that the conclusions over their role are true.

The paper also includes extensive analysis of metabolomics, with over a page of discussion and many Figures in the main document and supplementary. Figure 4 is impossible to read due to the small text size, and it is difficult to draw any significant conclusions from the data presented – substantial variations are seen between the different peptides, and between the same peptide in different media, raising the question as to whether these results have any relevance to where the peptides are actually meant to be used - in vivo. The authors need to demonstrate more convincingly that the differences in metabolic strategies can actually impact on how they peptides should be designed e.g. postulation In 489-492 should be supported by evidence that this effect is indeed seen.

The authors indicate in the accompanying Reporting Summary form that the exact sample size is specified for each experiment. This is not the case for many experiments: e.g. I could not find the group sizes of mice for the in vivo experiments in the text, figure legend, or experimental section. MIC and cytotoxicity results also do not specify n, or if biological replicates were conducted.

MIC assays: MIC assays were conducted in MHB, yet CA-MHB is now required by CLSI guidelines. Please justify why this is not used? No reference antibiotic MICs are included - these must be supplied, particularly for the strain used in the in vivo experiment. What is the n for the MIC results? MICs should be a minimum of two separate biological repetitions.

Haemolysis assay: 'No noticeable haemolytic activity at concentrations where they are effective antibacterials' is mentioned in the text, and experimental is provided, but no actual results are reported. This is critical, as the argument that the compounds are less toxic and have a better therapeutic index is moot if they are causing significant haemolysis at values close to the MIC. Both CC10 and CC50 values should be measured and reported.

In vivo efficacy: No n provided. Discussion and figure legend does not indicate at what timepoint animals are sacrificed for final cfu determination. For bacterial cfu experiments, a control group should be sacrificed at the time of antibiotic administration (e.g. 4h post infection in this case) to establish a baseline cfu count. With the current experiments it is impossible to say whether treatment is maintaining stasis, or if the cfu has increased/decreased from the initial infection. Is the cumulative 600 mg/kg dose of vancomycin needed to reach the modest level of efficacy observed, or can a lower dose also show a similar result?

Other issues:

Most of the figures are too small to read e.g. for Fig 1 one needs to be able to read the y-axis legends to see what is being reported in the heat maps, but the text is too small. Fig 2 - can't see structures of peptides in A/D/G/, and can't tell which curves belong to which compound in CD spectra.

Figure 3 - why are patch clamp results shown for different concentrations (varying from 5-50 μ M) for the different peptides, rather than the same concentration? There is no explanation in the text.

In 400: POL7080 clinical trial was stopped in May 2019 due to kidney damage - this should be mentioned.

In 491-422: discussion says that synergy with tobramycin could be useful approach, but in vivo data generated in this article shows no synergy, and indeed potential antagonism, so this statement is incongruous with the results.

In 466-469: implies that D-pleurocidin enantiomers are more stable due to proteolytic resistance. There is no direct evidence to support this claim - MS analysis of degradation in the 2 media should be done to confirm if this is true.

SI Figures: Why are there 6 different Figure 3s with separate legends? Within the 4th - 6th Figure 3's, what are the 4 different graphs for each of A, B, and C? The legend does not say. Figure 16, 19 and 24 have unreadable components due to overlap of text.

Figure 26: For synergy testing results, where is the efficacy result for D-pleurocidin alone?

Conclusions: The paper does not present any convincing data to overcome the current concern that systemic delivery of AMPs will never be possible due to insufficient therapeutic index. There is a marginal therapeutic index provided based on mammalian cell toxicity (up to 12-fold), but many existing in vitro studies of AMPs have shown greater selectivity. No TI over red blood cell haemolysis, a key concern of basic AMPs, is reported. Given the number of other more advanced AMP studies, claims of therapeutic potential really require evidence of an in vivo TI - needing at

least an acute maximum tolerated dose study, and preferably assessment of kidney damage after multiple days dosing, given nephrotoxicity is a proven liability of basic AMPs.

Overall, the paper contains some interesting design and analysis, but key points are lost in a sea of data, and the overall conclusions should be redrafted. A major revision is recommended.

Reviewer #3:

Remarks to the Author:

Background:

The authors have completed a highly detailed study which describes the development of multiple pleurocidin analogues and their testing as effective antibacterials both in silico and in vitro. They identified several analogues with improved spectrum and efficacy. They also investigated the mechanism of action and the impact of their analogues on bacterial metabolism. The most promising analogue, D- pleurocidin-KR was demonstrated to be effective in an in vivo mouse epidemic methicillin-resistant Staphylococcus aureus (EMRSA) lung model at 15mg/mg. In this work the authors make significant gains in enhancing the activity of their antimicrobial peptide analogues. This work would be of considerable interest to those in the field.

The authors have completed a significant body of work and compiled a well written manuscript with only minor typographical errors noted. The scale of the manuscript and detail included, with 26 supplementary figures does reduce the readability in early sections of this paper. While the authors could perhaps have considered two publications for this work, that should not detract from its recommended for publication with minor amendments.

Minor amendments

Line 25- based on information in the abstract it is unclear what is intended by "bacterial metabolic strategy", consider minor expansion of the definition for reader clarity.

Line 46- Consider amending to "...AMP with broad spectrum anti-bacterial activity that acts by damaging the plasma membrane.

Line 619- Murine EMRSA-15 lung infection model, insert details regarding the statistical tests used to assess the results.

Volcano plots – some of the text in volcano plots can be difficult to read, this may not be amendable.

Minor typographical errors noted

Line 84- "are" to "were"?

Line 89- define "POPG"

Line 101-define "POPE"

Line 151 define "DSSP" and "DISICL"

Line 170- define "MD" and "CD"

Line 205-define "DPhPG"

Line 222-define "DPhPE"

Line 302-define "DCFH-DA" fluorescence

Line 456-define "HDPs"

Line 467-define "FBS"

Line 533- missing " ° " in 37 C

Line 625- remove "either"

Reviewers' comments:

Reviewer #1 (Remarks to the Author):

In this paper, the authors describe modification strategies that generate pleurocidin analogues with substantially improved, broad spectrum, antibacterial properties, which are effective in murine models of bacterial lung infection. This is a well-written paper which merit publication. For the benefit of the readers, however, several points need clarifying and certain statements require further justification. There are given below.

1. Many researchers devoted to shorten the length of antimicrobial peptides [1, 2]. In this paper, the length of pleurocidin analogues are the same as pleurocidin. So, what are the advantages, compared with several other short peptides.

For reference [1], D-pleurocidin-KR is more potent than any of the shorter peptides based on a comparison with the parent peptide, pleurocidin. It is hard to make a direct comparison but in [1] Ple itself is not outperformed against S. mutans or S. sobrinus by any of the congeners when MICs are presented in µg/ml. Pm11 equals Ple against S. mutans and S. sobrinus and narrowly outperforms Ple against S. sanguinis. Pm11 is however about half the size of Ple and hence if MICs are converted to µM it is in fact inferior and indeed all the Ple congeners are less potent than Ple when molarity is considered. Pm11 is also cytotoxic at 4 x MIC.

For reference [2] the authors do not actually report the MIC for pleurocidin in their test conditions and hence it is not possible to easily or accurately compare the MICs of the shortened peptides. The authors merely state that "significant killing activity ... was only retained by Plc-2 and Plc-4 as compared to the parent peptide". The quoted MIC for Plc-2 E. coli ATCC-35218 of 4.0 µM in [2] compares with an MIC of 0.37 µM for E. coli NCTC 12923 in the present study. It is hard to argue then that any shortening of the pleurocidin sequence improves potency.

In contrast, in the present study including the -KR modification can provide up to a four-fold improvement in potency (depending on strains and conditions) while the D-amino acid modification further enhances potency such that in some cases there is a sixteen-fold difference in potency between the new analogue and the parent pleurocidin. We have added a line that briefly summarises these comparisons (line 420-422).

2. HEK293 is a cell line derived from human embryonic kidney cells. Hela cell line is derived from cervical cancer cells. However, the authors are focused on the bacteria which cause the lung infection and the Murine EMRSA-15 lung infection model is used in this paper. So, maybe lung cell line is suitable in cytotoxicity assay.

Despite restrictions imposed by closing our laboratories in response to Covid-19 we are pleased to report we have been able to do these suggested assays which, in hindsight, we agree are more appropriate than the data we presented with HEK293 and HeLa cells. With collaborators in Hong Kong (new authors Rico Man and Jenny Lam), we have conducted XTT assays using Calu-3 (bronchial epithelial) and A549 (type 2 pneumocyte) giving us coverage of different types of lung epithelial cells. The new data are presented as an addition to Table 2 and the same trends for the three D-amino acid analogues were observed. Accordingly, we have replaced the SI data for HEK293/ P. aeruginosa with SI for A549 / EMRSA-15.

[1] Zhang M, Wei W, Sun Y, Jiang X, Ying X, Tao R, et al. Pleurocidin congeners demonstrate activity against Streptococcus and low toxicity on gingival fibroblasts. Arch Oral Biol. 2016;70:79-87.

[2] Souza AL, Diaz-Dellavalle P, Cabrera A, Larranaga P, Dalla-Rizza M, De-Simone SG. Antimicrobial activity of pleurocidin is retained in Plc-2, a C-terminal 12-amino acid fragment. Peptides. 2013;45:78-84.

Reviewer #2 (Remarks to the Author):

This article describes the design, synthesis and characterisation of antimicrobial peptides (AMPs), analogues of the linear basic peptide pleurocidin.

The introduction and background is well written and clear. As the authors indicate, AMPs have been the subject of intense research over the past 30 years, yet none have successfully navigated a pathway to approval for systemic use as an antibiotic.

The authors rely on extensive membrane-based molecular dynamics simulations of two modified pleurocidin derivatives – one where Val residues have been modified to alanines to increase flexibility, and the other where Lys residues are altered to Arg. They also prepared all-D amino acid versions of the peptides, designed to improve proteolytic stability. However, one of the key conclusions from these studies is that the protonation state of the His residues plays a key role in how they act. It is surprising that these residues were not also replaced e.g. with permanently charged residues, to validate that the conclusions over their role are true.

The reviewer is correct that the role of the histidine residues seems critical in determining activity, in particular selectivity. It is of interest to explore this further and we are doing so in follow on work that has recently commenced. However, such an approach is beyond the scope of the present study for two reasons. First, the present study has a specific focus on conformational flexibility and describing the two modifications that achieve this is already a substantial task. Second, replacing the histidines with permanently charged residues is practically impossible to do without altering other properties – notably the hydrogen bonding capabilities and the position and flexibility of the side-chain. Presenting such an approach would require an in-depth characterisation of the effects appropriate for a separate study.

The paper also includes extensive analysis of metabolomics, with over a page of discussion and many Figures in the main document and supplementary. Figure 4 is impossible to read due to the small text size, and it is difficult to draw any significant conclusions from the data presented – substantial variations are seen between the different peptides, and between the same peptide in different media, raising the question as to whether these results have any relevance to where the peptides are actually meant to be used - in vivo. The authors need to demonstrate more convincingly that the differences in metabolic strategies can actually impact on how they peptides should be designed e.g. postulation In 489-492 should be supported by evidence that this effect is indeed seen.

The font size has been increased in all the Volcano plots in Figure 4.

We apologise if the message from the metabolomics experiments is not clear since we consider the information presented in these experiments to be substantial and the conclusions significant since they: 1) are based on consistent observations across three techniques (liquid state NMR, HR-MAS NMR and DCFH-DA fluorescence) and; 2) explain the critical differences in potency observed between analogues when tested in either MHB or RPMI. As discussed in the manuscript, and below in response to the reviewers point regarding CA-MHB, the point of doing the testing (and metabolomics) in these two different media is to frame exactly the question as to the relevance of these media to the in vitro setting. Critically for EMRSA-15, since its susceptibility to D-pleurocidin differs dramatically between MHB and RPMI (but that of D-pleurocidin-KR does not), the question is whether these bacteria use fermentation as a metabolic strategy in the lung infection setting? Although answering this question is beyond the scope of the present study, as discussed (lines 491-501), there is evidence that fermentation is not used by S. aureus during lung infection. This is perhaps surprising since glucose is typically abundant in such settings and is consumed avidly in vitro conditions. Ref #47 uses gene

expression analysis however and does not consider the activity of the enzymes involved nor measure whether fermentation products are produced. Consequently, until this question is conclusively resolved, the study design presented offers a means of being confident that a selected hit or lead will remain effective in the face of differing EMRSA-15 metabolism. We have added some text in the discussion and revised the final sentence of this section to clarify these points – notably highlighting that glucose is available in the infection setting (lines 498-500) and being more explicit about what our study design provides (lines 493-496 and 503-505).

The authors indicate in the accompanying Reporting Summary form that the exact sample size is specified for each experiment. This is not the case for many experiments: e.g. I could not find the group sizes of mice for the in vivo experiments in the text, figure legend, or experimental section. MIC and cytotoxicity results also do not specify n, or if biological replicates were conducted.

Apologies if this information did not reach the reviewer but, as mentioned at the point of submission, the Reporting Summary form indicates that this information will be in the revised version. We have been pleased to rectify this in the present revision.

MIC and cytotoxicity study data are all from three biological replicate experiments and this is made explicit in the materials and methods.

Group sizes for in vivo experiments are presented in the materials and methods.

DCFH-DA has been replotted to show all data points. In the interim we had conducted further replicate experiments and now all data points that we have for these conditions are shown (affects Figure 4 where $n = 24$ (7×3) but not Supp. Fig. 22 where $n = 6$ (2×3)).

Further, in the supplementary section, box plots have been revised to show individual data points (Supp. Fig. 12, 13, 15, 16, 20, 21, 23, 24).

MIC assays: MIC assays were conducted in MHB, yet CA-MHB is now required by CLSI guidelines. Please justify why this is not used? No reference antibiotic MICs are included - these must be supplied, particularly for the strain used in the in vivo experiment. What is the n for the MIC results? MICs should be a minimum of two separate biological repetitions.

While the CLSI do indeed recommend use of CA-MHB for general antibacterial susceptibility testing, they have not provided guidelines for testing AMPs. The key reference in the AMP susceptibility testing field is a Nature Protocol from 2008, reference #25 (which we cite in the materials and methods and also results), which has been cited 3087 times and guides most research on AMPs. This reference states "Please note however that cation-adjusted MHB should not be used when testing the activity of cationic antimicrobial peptides, as the presence of Ca^{2+} and Mg^{2+} ions causes a substantial inhibition of the cationic peptides' activity". The lack of calcium and magnesium ions in the media will therefore certainly impact on susceptibility but it is far from the only factor as our data obtained in RPMI show (where the calcium and magnesium ion concentration is comparable to that in CA-MHB). Elsewhere we have started to consider the role of individual components in determining species specific outcomes of AMP susceptibility testing (manuscript in revision with ACS Infectious Diseases) and, as discussed above, we have debated the extent to which different media are relevant to the in vivo setting in the manuscript discussion (lines 468-473). As mentioned in the manuscript, the use of both MHB and CA-MHB have been criticised recently and there is not yet any susceptibility testing media that are proven to predict in vivo performance. The comparison between MHB and RPMI therefore is relevant to researchers familiar with AMP research but also provides a new perspective on how AMP susceptibility testing may influence hit selection.

Reference MIC data, using relevant comparator antibiotics, is now provided for the strains used in the basic screening of activity and for the EMRSA-15 strain used in the in vivo study (Supp. Table 2). All MICs are generated from at least three biological replicate experiments. We have updated the manuscript to reflect this.

Haemolysis assay: 'No noticeable haemolytic activity at concentrations where they are effective antibacterials' is mentioned in the text, and experimental is provided, but no actual results are reported. This is critical, as the argument that the compounds are less toxic and have a better therapeutic index is moot if they are causing significant haemolysis at values close to the MIC. Both CC10 and CC50 values should be measured and reported.

The reviewer is absolutely right, and we apologise for not including this data. We did not see sufficient haemolysis to calculate HC10 and HC50 values for all peptides. Only pleurocidin and D-pleurocidin-KR attain 50% haemolysis (relative to Triton X100 control) at the highest concentration tested 400 µg/ml. The data is now presented in the supplementary material (Supp. Fig. 13) and none of the peptides cause significant haemolysis close to the MIC. In the case of D-pleurocidin-KR the MICs are generally below 4 µg/ml and 10% haemolysis is achieved only above 64 µg/ml.

In vivo efficacy: No n provided. Discussion and figure legend does not indicate at what timepoint animals are sacrificed for final cfu determination. For bacterial cfu experiments, a control group should be sacrificed at the time of antibiotic administration (e.g. 4h post infection in this case) to establish a baseline cfu count. With the current experiments it is impossible to say whether treatment is maintaining stasis, or if the cfu has increased/decreased from the initial infection. Is the cumulative 600 mg/kg dose of vancomycin needed to reach the modest level of efficacy observed, or can a lower dose also show a similar result?

Apologies, n is now given in the materials and methods. Data presented are an aggregate of two independent repeats, each with control groups (n = 6) and peptide treatment groups (n = 4).

The timepoint for sacrifice (48h) was provided in the materials & methods but is now also included in the legend for Figure 5.

We have done the requested experiment taking a control group and sacrificing at 4h. We observe a mean of $4.26 \pm 0.24 \log_{10}$ CFU/ml which is comparable to the mean of $4.43 \pm 0.97 \log_{10}$ CFU/ml shown for 48h. As such the CFU is stable across the time course of the experiment and a statement to this effect is now included (line 371).

We have previously shown (unpublished results) that when used against USA300, 200 mg/kg vancomycin is required to get a similar reduction in CFU counts (1-1.5 log reduction). When we set up the present study for multiple dosing, we used this dose to see if we could get a greater reduction, however we saw a similar response. Although the in vitro MICs are similar, this is a different strain to the one used in the present study and so we have included the statement "(a similar reduction may be achieved with 200 mg/kg vancomycin, data not shown)" (line 376).

Other issues:

Most of the figures are too small to read e.g. for Fig 1 one needs to be able to read the y-axis legends to see what is being reported in the heat maps, but the text is too small. Fig 2 – can't see structures of peptides in A/D/G/, and can't tell which curves belong to which compound in CD spectra. Figure 3 – why are patch clamp results shown for different concentrations (varying from 5-50 µM) for the different peptides, rather than the same concentration? There is no explanation in the text.

Apologies for the legibility of the figures. We have increased the font size the axes and tick labels for all panels in Figure 1 and used bold for residue names. The size of Figure 2 has been increased. We have changed the key and thickened the curves in the CD spectra so these can be more easily discerned (and increased the font size for the tick and axes labels). The font size for tick labels in panels B, E and H have also been increased. Tick and axes labels have increased for Figure 3 panels A, C, E, G, I, K, O and P and residues labels are in bold. We have increased the font size for the metabolite labels in Figure 4.

Apologies for not including details of the patch-clamp study design in the text. This would ordinarily be in the materials and methods but to save space we referred only to previous work where this approach has been described. We perform a titration to find the lowest peptide concentration that induces detectable membrane activity. We have now included this explanation in the figure legend.

In 400: POL7080 clinical trial was stopped in May 2019 due to kidney damage – this should be mentioned.

We thank the reviewer for alerting us to this and we have modified the section accordingly – we understand POL7080/murepavadin is now being developed for inhaled delivery and hence it has been included as one of three peptides discontinued for i.v. delivery in the discussion (line 394).

In 491-422: discussion says that synergy with tobramycin could be useful approach, but in vivo data generated in this article shows no synergy, and indeed potential antagonism, so this statement is incongruous with the results.

*We do not agree that there is any evidence of potential antagonism (the highest dose - 15 mg/kg D-pleurocidin-KR - does not lead to any impairment of the effect of 25 mg/kg tobramycin). However, on reflection we consider presenting the data in the original supplementary figure to be premature. This is because we observe a possible “sweet spot” effect that we have also now recently observed in analogous studies in *Galleria mellonella* (wax moth) models which suggest a more complex dose dependency in some infection settings. This should be studied and understood in more detail before being communicated. The data do not add substantially to the manuscript and we have hence removed this from the revision. The in vitro data remain as we consider this will remain a likely route of exploiting, in particular, new leads that will follow the hits presented in the present study. However, they have been moved to the supplementary.*

In 466-469: implies that D-pleurocidin enantiomers are more stable due to proteolytic resistance. There is no direct evidence to support this claim – MS analysis of degradation in the 2 media should be done to confirm if this is true.

While we agree with the reviewer that we have not tested this property here, it is common practice to use all D-enantiomers in place of the L-enantiomer to protect from proteolysis. This has also been explicitly demonstrated in at least two previous studies for pleurocidin/D-pleurocidin and we now make reference to one of these studies in the results section (Jung et al, 2007). Since we have shown no difference in conformation between the all-L and all-D enantiomers (Supp. Fig. 10) and do not see any noticeable difference between potencies in MHB there is no evidence to support a different mechanism of action between all-L and all-D enantiomers (as now stated on line 250). There are however substantial differences in activity between all-L and all-D enantiomers in RPMI and since the difference in susceptibility to proteolysis is known and we know that proteases are added to this media (in the FBS) it seems a reasonable inference that peptide stability underpins these differences.

SI Figures: Why are there 6 different Figure 3s with separate legends? Within the 4th - 6th Figure 3's, what are the 4 different graphs for each of A, B, and C? The legend does not say. Figure 16, 19 and 24 have unreadable components due to overlap of text.

Each panel that constitutes "Secondary structure analysis of pleurocidin peptides from MD simulations" and, as such, is the same data but analysed in different ways. For this reason, we considered it appropriate to consider these all as one figure. However, to provide further clarity we have renumbered the supplementary figures and added further detail to the legends. The DISICL analysis is presented with a separate panel for each peptide. This has been clarified in the figure legend. The overlap in the metabolite annotation in Volcano plots is a challenge for those metabolites that do not change much or where there is low significance as they all cluster in the same place. Since they are below the significance threshold, we have elected to remove these labels.

Figure 26: For synergy testing results, where is the efficacy result for D-pleurocidin alone?

Based on our in vitro testing we did not expect an effect for the peptide alone and considered it unethical to perform dose-ranging experiments to confirm this. Instead our experimental design was based on reducing the effective tobramycin dose to a point where any significant improvement in the presence of D-pleurocidin-KR would suggest a synergistic effect. As mentioned above however, we have now removed this data from the SI.

Conclusions: The paper does not present any convincing data to overcome the current concern that systemic delivery of AMPs will never be possible due to insufficient therapeutic index. There is a marginal therapeutic index provided based on mammalian cell toxicity (up to 12-fold), but many existing in vitro studies of AMPs have shown greater selectivity. No TI over red blood cell haemolysis, a key concern of basic AMPs, is reported. Given the number of other more advanced AMP studies, claims of therapeutic potential really require evidence of an in vivo TI – needing at least an acute maximum tolerated dose study, and preferably assessment of kidney damage after multiple days dosing, given nephrotoxicity is a proven liability of basic AMPs.

This concern should be largely addressed by the inclusion of the haemolysis data that shows that, depending on whether MIC data in MHB or RPMI (5% FBS) is considered, the TI over red blood cell haemolysis will be at least 100-fold and may be more than 400-fold (line added 261-262). The TI over epithelial cell cytotoxicity is hence more pertinent and is included in Table 2 (now with additional data for A549). Again, depending on the MIC data being considered the therapeutic index varies between 28- and 113-fold.

Further, we have been fortunate to take advantage of NIAID's suite of pre-clinical services for a maximum tolerated dose (MTD) study. We have not yet moved to dosing beyond 48 hours as this would require a PK study which is beyond the scope of the present communication. However, D-pleurocidin-KR was tolerated at 15 mg/kg/24h when administered i.v. to ICR mice. Similar reductions in lung CFU are shown here with 15 and 1.5 mg/kg/48h and hence the in vivo TI is approximately 20-fold (line added 379-380) and has been achieved with no optimisation of dosing. This can be compared with e.g. a recent study published in Nature Communications (<https://doi.org/10.1038/s41467-020-16950-x>) where the difference between effective doses and the threshold for adverse effects was 10-fold.

Overall, the paper contains some interesting design and analysis, but key points are lost in a sea of data, and the overall conclusions should be redrafted. A major revision is recommended.

We trust that in addressing the comments made above and making substantial revisions to the manuscript, including our concluding remarks, the importance of the approach and findings are now readily apparent.

Reviewer #3 (Remarks to the Author):

Background:

The authors have completed a highly detailed study which describes the development of multiple pleurocidin analogues and their testing as effective antibacterials both in silico and in vitro. They identified several analogues with improved spectrum and efficacy. They also investigated the mechanism of action and the impact of their analogues on bacterial metabolism. The most promising analogue, D- pleurocidin-KR was demonstrated to be effective in an in vivo mouse epidemic methicillin-resistant Staphylococcus aureus (EMRSA) lung model at 15mg/mg. In this work the authors make significant gains in enhancing the activity of their antimicrobial peptide analogues. This work would be of considerable interest to those in the field.

The authors have completed a significant body of work and compiled a well written manuscript with only minor typographical errors noted. The scale of the manuscript and detail included, with 26 supplementary figures does reduce the readability in early sections of this paper. While the authors could perhaps have considered two publications for this work, that should not detract from its recommended for publication with minor amendments.

We thank the reviewer for the kind comments and reflect that indeed we did consider drafting two manuscripts. We opted for a single manuscript as we consider the relationship between the biophysical and metabolomic studies will be of particular interest while the animal studies enable the impact of such approaches to be fully appreciated.

Minor amendments

Line 25- based on information in the abstract it is unclear what is intended by “bacterial metabolic strategy”, consider minor expansion of the definition for reader clarity.

Thank you for highlighting this apparent confusion. We have split the sentence into two to better show what we mean by this property: “Increasing peptide-lipid intermolecular hydrogen bonding capabilities enhances conformational flexibility, associated with membrane translocation, but also membrane damage and potency, most notably against Gram-positive bacteria. This negates their ability to metabolically adapt to the AMP threat.”

Line 46- Consider amending to “...AMP with broad spectrum anti-bacterial activity that acts by damaging the plasma membrane.

We agree that this simplifies the sentence, the intended nuance may not be appreciated by most readers and the counterpoint that pleurocidin acts on intracellular targets is presented soon afterwards so the meaning of the passage is unaffected. We have therefore adopted this suggestion.

Line 619- Murine EMRSA-15 lung infection model, insert details regarding the statistical tests used to assess the results.

Apologies for this oversight. We have added a line at the end of the section indicating we used a One-way ANOVA with Dunnett’s multiple comparisons test.

Volcano plots – some of the text in volcano plots can be difficult to read, this may not be amendable.

We agree. We have increased the font size of the metabolite labels in the Volcano plots to make these easier to read both in Figure 4 and in the various supplementary figures.

Minor typographical errors noted – *Many thanks for picking these up!*

Line 84- "are" to "were"? *Done*

Line 89- define "POPG" *Done*

Line 101-define "POPE" *Done*

Line 151 define "DSSP" and "DISICL" *Done*

Line 170- define "MD" and "CD" *Done*

Line 205-define "DPhPG" *Done*

Line 222-define "DPhPE" *Done*

Line 302-define "DCFH-DA" fluorescence *Done*

Line 456-define "HDPs" *Done*

Line 467-define "FBS" *Defined in line 258.*

Line 533- missing " ° " in 37 C *Thank you!*

Line 625- remove "either" *Thank you!*

REVIEWERS' COMMENTS:

Reviewer #1 (Remarks to the Author):

In the revise paper, the research group has summarise the comparisons of several other short peptides. The research group also has conducted XTT assays using Calu-3 (bronchial epithelial) and A549 (type 2 pneumocyte) giving us coverage of different types of lung epithelial cells. This is a well-written paper that meets the requirements for publication.

Reviewer #2 (Remarks to the Author):

The authors have substantively addressed the reviewer concerns.

However, a couple of interpretations should be corrected:

In 261: For haemolysis, HC10 is commonly used to describe the acceptable threshold for calculating a therapeutic index, not HC50 (e.g. see use in doi: 10.1038/srep09761, doi: 10.1021/ja500367u, doi.org/10.1038/s41467-020-16950-x) and can be calculated based on Fig S13 as all peptides achieve this level. For DPKR the HC10 appears to be approximately 80 ug/mL, so the TI is around 40 based on an average MIC of 2 ug/mL, not 400.

In 274-277 Incorrect interpretation of combination antibiotic activity. Table S1 does not show synergy. FICI values >0.5 and <1 are additive, not synergistic. FICI definitions: synergy, FICI of ≤ 0.5 ; additivity, FICI of >0.5 to ≤ 1 ; no interaction (indifference), FICI of >1 to ≤ 4 ; antagonism, FICI of >4 . v see DOI: 10.1128/AAC.00497-10 and elsewhere

In 376, please clarify further the sentence to ""(a similar reduction may be achieved with 200 mg/kg vancomycin, as found in a previous study using a USA300 strain: data not shown)""

Figure 1 appears identical in the original and revised versions, I cannot see that the stated changes have been made.

Reviewers' comments:

Reviewer #1 (Remarks to the Author):

In the revise paper, the research group has summarised the comparisons of several other short peptides. The research group also has conducted XTT assays using Calu-3 (bronchial epithelial) and A549 (type 2 pneumocyte) giving us coverage of different types of lung epithelial cells. This is a well-written paper that meets the requirements for publication.

Thank you very much.

Reviewer #2 (Remarks to the Author):

The authors have substantively addressed the reviewer concerns. However, a couple of interpretations should be corrected:

In 261: For haemolysis, HC10 is commonly used to describe the acceptable threshold for calculating a therapeutic index, not HC50 (e.g. see use in doi: 10.1038/srep09761, doi: 10.1021/ja500367u, doi.org/10.1038/s41467-020-16950-x) and can be calculated based on Fig S13 as all peptides achieve this level. For DPKR the HC10 appears to be approximately 80 ug/mL, so the TI is around 40 based on an average MIC of 2 ug/mL, not 400.

We were focussing on the MIC for EMRSA-15 in RPMI (0.5 mg/ml) when making this statement since this is the isolate selected for the in vivo studies. We acknowledge this is selectively optimistic and have hence revised this sentence to read "...and the in vitro therapeutic index over red blood cell haemolysis for the majority of isolates ranges between 18- and 150-fold for D-pleurocidin-KR (10% haemolysis achieved with approx. 75 µg/ml)." This considers all isolates bar P. aeruginosa and E. faecalis.

In 274-277 Incorrect interpretation of combination antibiotic activity. Table S1 does not show synergy. FICI values >0.5 and <1 are additive, not synergistic. FICI definitions: synergy, FICI of ≤0.5; additivity, FICI of >0.5 to ≤1; no interaction (indifference), FICI of >1 to ≤4; antagonism, FICI of >4. v see DOI: 10.1128/AAC.00497-10 and elsewhere.

The reviewer is absolutely correct that historically (key papers in 2000 and 2003)^{1,2} 0.5 has been the widely accepted threshold for synergy. However, this is a conservative value that has been adopted due to uncertainty regarding the reliability of dilution in checkerboard assays when the true, mathematical threshold is 1.0. There is therefore the risk that all modest synergy will be ignored with this approach – one might reduce the amount of each of two antibiotics by more than two-thirds and still not meet this threshold. More recently (2017) Fratini et al addressed this limitation by arguing that, if the MIC used in the FIC calculation is determined simultaneously in the same microarray plate, the FIC calculated will be much more robust to variation in dilution. We consider it important that even modest synergy ("modest" added to line 276) should be reported and have added a line of explanation in the methods section of the manuscript (548-550) as well as a more detailed explanation in the legend to the Supplementary Table (now #2).

In 376, please clarify further the sentence to ""(a similar reduction may be achieved with 200 mg/kg vancomycin, as found in a previous study using a USA300 strain: data not shown)".

This statement was included to indicate that the 600 mg/kg dose used in the present study is likely a little more than is required. This dose was selected on the basis of previous experiments conducted with the same model albeit with a different strain of S. aureus and with vancomycin delivered intra

peritoneal. *These experiments are described in a published PhD thesis from co-author BGO'S. We have changed the statement to read "a similar reduction may be achieved in this model with lower doses, c.f. 200 mg/kg vancomycin i.p. gives a comparable reduction in S. aureus USA300 CFU³⁷".*

Figure 1 appears identical in the original and revised versions, I cannot see that the stated changes have been made.

Thank you for pointing this out. The correct figure has been inserted which also now includes lower case panel labels.

References:

1. EUCAST, 2000. Terminology relating to methods for the determination of susceptibility of bacteria to antimicrobial agents. *Clin. Microbiol. Infect.* 6,503–508, <http://dx.doi.org/10.1046/j.1469-0691.2000.00149.x>.
2. Odds, F.C., 2003. Synergy, antagonism, and what the chequerboard puts between them. *J. Antimicrob. Chemother.* 52, 1, <http://dx.doi.org/10.1093/jac/dkg301>.